# END-TO-END LEARNING OF PROBABILISTIC HIERARCHIES ON GRAPHS

**Daniel Zügner, Bertrand Charpentier, Sascha Geringer, Morgane Ayle,**
**Stephan Günnemann**
Technical University of Munich
`{zuegnerd, charpent, geringer, ayle, guennemann}@in.tum.de`

## ABSTRACT

We propose a novel probabilistic model over hierarchies on graphs obtained by continuous relaxation of tree-based hierarchies. We draw connections to Markov chain theory, enabling us to perform hierarchical clustering by efficient end-to-end optimization of relaxed versions of quality metrics such as Dasgupta cost or Tree-Sampling Divergence (TSD). We show that our model learns rich, high-quality hierarchies present in 11 real world graphs, including a large graph with 2.3M nodes. Our model consistently outperforms recent as well as strong traditional baselines such as average linkage. Our model also obtains strong results on link prediction despite not being trained on this task, highlighting the quality of the hierarchies discovered by our model.

## 1 INTRODUCTION

Clustering is a fundamental problem in unsupervised learning, both in theory and practice. In contrast to traditional 'flat' clustering, hierarchical clustering brings several advantages. It enables us to naturally analyze (Zhao & Karypis, 2002) and visualize (Himberg et al., 2004) a given dataset on different scales. Further, different downstream tasks may require different granularities. Hierarchical clustering lets us pick the desired granularity after learning. It can also be used for personalized recommendation (Zhang et al., 2014) or to solve record linkage tasks (Wick et al., 2012). In an influential paper, Eisen et al. (1998) perform hierarchical clustering on gene expression data and argue that it could be used to discover the yet-unknown meaning of certain genes, which could lead to advances in medicine. Besides traditional vector data, real-world graphs are often scale-free and hierarchically organized (Ravasz & Barabási, 2003; Barabási & Pósfai, 2016). This makes them interesting candidates for uncovering the underlying hierarchy via hierarchical clustering. Thus, the hierarchical clustering method we introduce is particularly useful for analyzing scale-free real-world networks such as Web graphs, citation networks, flight networks, or biological networks.

**Related work.** We can group hierarchical clustering algorithms into discrete and continuous optimization methods. Discrete optimization methods can themselves be decomposed into agglomerative approaches and divisive approaches. Agglomerative methods – like the famous linkage algorithms (Gower & Ross, 1969; Jardine & Sibson, 1968; Bonald et al., 2018; Charpentier, 2019) – follow a bottom-up approach and iteratively aggregate the two most similar clusters w.r.t. a given similarity measure. Divisive methods follow a top-down approach and recursively split larger clusters into smaller ones using, e.g., k-means (Steinbach et al., 2000) or spectral clustering (Charikar & Chatziafratis, 2016). While Moseley & Wang (2017); Kamvar et al. (2002) have shown connections with explicit objective functions, those methods are mainly based on heuristics to avoid exhaustive combinatorial optimization.

On the other hand, continuous optimization methods circumvent this issue by **(i)** relaxing the discrete tree structure, **(ii)** defining an explicit quality metric for hierarchies, and **(iii)** optimizing it using gradient-based optimizers. Chierchia & Perret (2019); Monath et al. (2017; 2019); Chami et al. (2020) focus on minimizing the Dasgupta cost (Dasgupta, 2016) (see Eq. (1)) to fit an ultrametric, hyperbolic embeddings, or a probabilistic model over cluster assignments. Monath et al. (2017) learns a probabilistic assignment of the samples to leaves of a tree with a fixed binary structure. The method of Monath et al. (2019) requires approximation for the computation of lowest common ancestors

(LCA) probabilities and regularization losses. Beyond the Dasgupta cost, Charpentier & Bonald (2019) propose the Tree Sampling Divergence (TSD), a quality metric for hierarchies on graphs. Both metrics have the benefit to be internal metrics, i.e., they do not require ground truth information to evaluate the quality of a hierarchy. Most of the aforementioned methods focus on vector data.

**Contributions.** We **(1)** propose a probabilistic model over hierarchies via continuous relaxation of a tree's parent assignment matrices. We **(2)** theoretically analyze the model by drawing connections to absorbing Markov chains, which allows **(3)** efficient and exact computation of relevant quantities (e.g., LCA probabilities). This enables us to **(4)** learn hierarchies on graphs by efficient, end-to-end optimization of relaxed versions of quality metrics such as Dasgupta cost and the TSD score. Our extensive experimental evaluation on 11 real-world graphs, including a massive graph with more than 2M nodes and 60M edges, highlights the effectiveness of our Flexible Probabilistic Hierarchy model **(FPH)**. It outperforms all baselines, traditional and recent, on the quality of the learned hierarchies measured both by TSD and Dasgupta cost. Remarkably, FPH also performs competitively on link prediction despite not being trained on this task.

## 2 HIERARCHICAL GRAPH CLUSTERING PRELIMINARIES

We define a graph $\mathcal{G} = (E, V)$ with $n$ nodes and $m$ undirected edges. We let $w(v_i, v_j)$ be equal to the weight of edge $(v_i, v_j)$ if they are connected and to 0 otherwise, and $w(v_i) = \sum_j w(v_i, v_j)$ to be equal to the weight of node $v_i$. This allows us to define the *edge distribution* $P(v_i, v_j) \propto w(v_i, v_j)$ normalized over all edges s.t. $\sum_{ij} P(v_i, v_j) = 1$ and the *node distribution* $P(v_i) \propto w(v_i)$ s.t. $\sum_i P(v_i) = 1$, which are later needed to compute the quality metrics of learned hierarchies.

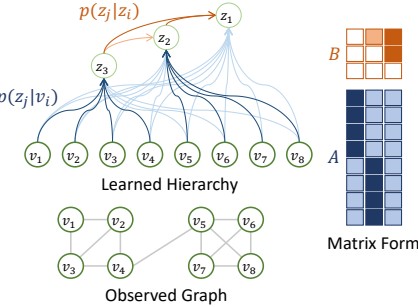

Figure 1: Model overview.

**Discrete hierarchical clustering.** We define a *hierarchical clustering* $\hat{\mathcal{T}}$ of a graph $\mathcal{G}$ as a tree structured partitioning of the nodes $V$. We denote the nodes of the tree $\hat{\mathcal{T}}$ with $n' \in \mathbb{N}$ internal nodes by $\{x_1, ..., x_{n+n'}\}$. The tree nodes can be decomposed into the leaf nodes $\{v_1, ..., v_n\}$ corresponding to the nodes of the input graph, and the internal nodes $\{z_1, ..., z_{n'}\}$ representing clusters of leaf nodes; overall $\{x_1, ..., x_{n+n'}\} = \{v_1, ..., v_n\} \cup \{z_1, ..., z_{n'}\}$. Since any tree is a directed acyclic graph (DAG), we impose w.l.o.g. a topological ordering on the internal nodes of the tree in the remainder of this work. That is, $z_i$ cannot be a parent of $z_j$ if $i < j$; $z_{n'}$ denotes the root node. We can uniquely define a tree as $\hat{\mathcal{T}} = (\hat{A}, \hat{B})$ with two adjacency matrices, $\hat{A} \in \{0, 1\}^{n \times n'}$ and $\hat{B} \in \{0, 1\}^{n' \times n'}$. $\hat{A}$ describes the connections from the leaves to the internal nodes, i.e. $\hat{A}_{i,k} = 1$ if internal node $z_k$ is the parent of leaf node $v_i$. $\hat{B}$ describes the connections between the internal nodes, i.e. $B_{k,l} = 1$ if internal node $z_l$ is the parent of internal node $z_k$. Since in a tree any node has *exactly* one parent, $\hat{A}$ is row-stochastic and $\hat{B}$ is row-stochastic; except its last row $\hat{B}_{n'}$ corresponding to the root, whose sum is 0. We denote by $v_i \wedge v_j$ the internal node which is the lowest common ancestor (LCA) of $v_i, v_j$ in the tree. The maximum possible number of internal nodes is $n' = n - 1$.

**Probabilistic hierarchical clustering.** The central idea of FPH is to define a *probabilistic* tree $\mathcal{T}$ via continuous relaxation of the binary-valued adjacency matrices $\hat{A}$, $\hat{B}$. I.e., we define $A$, $B$ such that $A \in [0, 1]^{n \times n'}$ and $B \in [0, 1]^{n' \times n'}$ while keeping the row-stochasticity constraints, i.e., $\sum_j^{n'} A_{i,j} = 1$ for $1 \leq i \leq n$, $\sum_l^{n'} B_{k,l} = 1$, $1 \leq k < n'$, and for the root node we have $\sum_l^{n'} B_{n',l} = 0$. This describes a probabilistic hierarchy; the probability of an internal node $z_j$ to be a parent of a leaf or an internal node is $A_{ij} := p(z_j|v_i)$ or $B_{ij} := p(z_j|z_i)$, respectively. See Fig. 1 for a visualization. Our key technical contribution are efficient closed-form expressions for the lowest common ancestor probabilities that arise when sampling discrete trees from the probabilistic one.

### 2.1 INTERNAL METRICS FOR HIERARCHICAL CLUSTERING

Internal metrics are commonly used to evaluate clustering methods, and they have the important advantage of not requiring external labels. For hierarchical clustering on graphs, two internal metrics have been proposed to assess the quality of a discrete hierarchy $\hat{\mathcal{T}}$ given an input graph $\mathcal{G}$: Dasgupta

cost (**Das.**) (Dasgupta, 2016), and Tree-Sampling Divergence (**TSD**) (Charpentier & Bonald, 2019). While the Dasgupta cost can be used for graphs and vector data, TSD is tailored specifically to graphs.

$$\text{Das}(\hat{\mathcal{T}}) = \sum_{v_i, v_j \in V} P(v_i, v_j) \sum_z \mathbb{I}_{[z = v_i \wedge v_j]} c(z), \quad \text{TSD}(\hat{\mathcal{T}}) = \text{KL}(p(z) || q(z)) \tag{1}$$

For **Das.**, $c(z) = \sum_{v_i \in V} \mathbb{I}_{[z \in \text{anc}(v_i)]}$ denotes the number of leaves for which the internal node z is an ancestor. Intuitively, the cost is large if $v_i$ and $v_j$ are connected by an edge *and* their lowest common ancestor contains many leaves, i.e., is close to the root. Thus, Dasgupta favors leaves connected by an edge to have LCAs low in the hierarchy. For Das., lower scores indicate better hierarchies.

**TSD** defines two distributions over the internal nodes, $p(z) = \sum_{v_i, v_j} \mathbb{I}_{[z = v_i \wedge v_j]} P(v_i, v_j)$, induced by the edge distribution, and $q(z) = \sum_{v_i, v_j} \mathbb{I}_{[z = v_i \wedge v_j]} P(v_i) P(v_j)$, induced by the independent node distribution. Intuitively, we expect the graph distribution $p(z)$ (induced by edge sampling) and the null model distribution $q(z)$ (induced by independent node sampling) to differ significantly if the tree $\hat{\mathcal{T}}$ indeed represents the hierarchical structure of the graph, leading to high TSD scores.

**Continuous relaxation.** To enable end-to-end gradient-based optimization of Dasgupta or TSD, we replace the *discrete* hierarchy $\hat{\mathcal{T}} = (\hat{\boldsymbol{A}}, \hat{\boldsymbol{B}})$ with its *continuous* relaxation $\mathcal{T} = (\boldsymbol{A}, \boldsymbol{B})$. This leads to probabilistic parent assignments of leaves to internal nodes. By itself, this does not help us in hierarchical clustering as it is not clear how to compute the lowest common ancestor probabilities when sampling discrete hierarchies from the relaxed one, i.e., the probability that an internal node is the lowest common ancestor of two leaves. Our main theoretical contribution is to derive closed-form expressions for the lowest common ancestor probabilities in Sec. 3. This gives rise to continuous versions of TSD and Dasgupta (details in Appendix A.1), which we will refer to as Soft-TSD and Soft-Das., respectively. For the first time, this allows us to *directly* optimize for hierarchical clustering quality metrics in an end-to-end fashion instead of proxy losses or heuristic algorithms.

For optimization over hierarchies, we either minimize the Dasgupta cost or maximize TSD:

$$\min_{\boldsymbol{A}, \boldsymbol{B}} \text{Soft-Das}(\mathcal{T} = (\boldsymbol{A}, \boldsymbol{B})) \qquad \text{or} \qquad \max_{\boldsymbol{A}, \boldsymbol{B}} \text{Soft-TSD}(\mathcal{T} = (\boldsymbol{A}, \boldsymbol{B})), \tag{2}$$

where $\boldsymbol{A} \in [0, 1]^{n \times n'}, \boldsymbol{B} \in [0, 1]^{n' \times n'}$ are the continuous relaxations of the parent assignment matrices as described in Sec. 2. Note that for both metrics we recover the same score as their discrete versions in the case of a deterministic probabilistic model, i.e. when $\boldsymbol{A}, \boldsymbol{B}$ are binary-valued.

## 2.2 SAMPLING DISCRETE HIERARCHIES

Given the probability matrices $\boldsymbol{A}$ and $\boldsymbol{B}$ we can easily recover a discrete hierarchical clustering by sampling discrete matrices $\hat{\boldsymbol{A}}$ and $\hat{\boldsymbol{B}}$, which in turn describe a (discrete) tree $\mathcal{T}$. For each leaf and internal node we independently sample its parent from the categorical distribution described by the respective row in $\boldsymbol{A}$ or $\boldsymbol{B}$. As we show in Appendix A.2, this *tree-sampling* procedure (denoted $\hat{\mathcal{T}} = (\hat{\boldsymbol{A}}, \hat{\boldsymbol{B}}) \sim P_{\boldsymbol{A}, \boldsymbol{B}}(\mathcal{T})$) leads to valid tree hierarchies, and we can directly compute the probability of any discrete hierarchy given the probabilistic one. We denote probabilities associated with the tree-sampling perspective as $p^{(\mathcal{T})}(\cdot)$. Note that we can easily obtain a discrete tree given continuous $\boldsymbol{A}, \boldsymbol{B}$ in a deterministic way by selecting for each leaf and internal node its most likely parent.

## 3 EFFICIENT, DIFFERENTIABLE HIERARCHIES VIA MARKOV CHAINS

**Outline.** This section is organized as follows. The goal is to derive efficient, closed-form equations to compute (lowest common) ancestor probabilities which are consistent with the *tree-sampling procedure* explained in the previous section. For this, we draw connections to absorbing Markov chains in Sec. 3 and show in Secs. 3.1, 3.2 how to compute the desired quantities efficiently under this simplifying *Markov Chain perspective*. Then, we show that these equations under the Markov chain and the tree-sampling perspectives are equivalent, i.e., we do not introduce any error by the Markov chain assumptions. Finally, we show in Sec. 3.3 how to exploit the independence assumptions of the Markov chain to compute the (lowest common) ancestor probabilities efficiently and in a vectorized way.

We start by showing that the probabilistic model described in the previous section indeed defines an absorbing Markov chain. Intuitively, $\boldsymbol{A}$ and $\boldsymbol{B}$ can be interpreted as *transition matrices* from leaves to internal nodes and among internal nodes, respectively.

**Definition 1** (Tree Markov Chain). *Let $A \in [0,1]^{n \times n'}$, $\sum_{j=1}^{n'} A_{ij} = 1 \; \forall 1 \leq i \leq n$ and $B \in [0,1]^{n' \times n'}$, $\sum_{j=1}^{n'} B_{ij} = 1 \; \forall 1 \leq i < n'$, $\sum_{j=1}^{n'} B_{n'j} = 0$. We define a Markov chain $\mathcal{M} = (\mathcal{S} \cup \{\omega\}, T)$ with state set $\mathcal{S} \cup \{\omega\}$ and transition matrix $T$, where $\mathcal{S} = \{v_1, v_2, \ldots, v_n\} \cup \{z_1, z_2, \ldots, z_{n'}\}$. Further,*

$$T \in \mathbb{R}^{|\mathcal{S}|+1 \times |\mathcal{S}|+1} = \begin{bmatrix} Q & w \\ 0 & 1 \end{bmatrix}, Q \in \mathbb{R}^{|\mathcal{S}| \times |\mathcal{S}|} = \begin{bmatrix} 0 & A \\ 0 & B \end{bmatrix}$$

*is the transition matrix of the Markov chain $\mathcal{M}$ in canonical form, where $w \in \mathbb{R}^{|\mathcal{S}|} = (0, 0, \ldots, 0, 1)^T$ is the vector of transition probabilities to state $\omega$. Note that we add here an auxiliary state $\omega$, which acts as final absorbing state from the root.*

**Theorem 1.** *Let $\mathcal{M}$ be a tree Markov chain as defined in Definition 1. $\mathcal{M}$ is an acyclic absorbing Markov chain with $\omega$ being its only absorbing state. (See proof in App. A.3)*

Thus, our probabilistic hierarchy defined by $A$ and $B$ describes an absorbing and acyclic Markov chain. In our hierarchical clustering interpretation, a random walk starts at one of the leaves and first randomly transitions to the internal nodes based on $A$, followed by transitions among the internal nodes via $B$. Once the root is reached, the random walk is absorbed after one last step to $\omega$.

In the remainder, we denote the probabilities arising from the tree Markov-chain as $p^{(\mathcal{M})}(\cdot)$ to distinguish them from their tree-sampling counterparts $p^{(\mathcal{T})}(\cdot)$. In general, the two perspectives are not equal, e.g., when considering pairs of leaf nodes. Here, the Markov chain introduces an independence assumption of pairs of random walks. Under tree-sampling, two leaves' paths to the root are dependent in general, i.e., two paths traversing some internal node $z_k$ implies that all subsequent transitions are identical. While Markov-chain probabilities $p^{(\mathcal{M})}(\cdot)$ often have efficient and analytical solutions grounded in Markov chain theory, it is unclear a-priori how to leverage them for hierarchical clustering. In the following, we relate central tree-sampling probabilities $p^{(\mathcal{T})}(\cdot)$, i.e., (lowest common) ancestor probabilities, to their Markov chain counterparts $p^{(\mathcal{M})}(\cdot)$.

### 3.1 Ancestor probabilities

First, we want to compute the probability $p_{\text{anc}}^{(\mathcal{T})}(z_k | v_j)$ of an internal node $z_k$ being an ancestor of leaf $v_j$ under the row-wise tree-sampling procedure (see Sec. 2.2). We call this an *ancestor* probability.

**Theorem 2.** *Let $\mathcal{T} = (\hat{A}, \hat{B}) \sim P_{A,B}(\mathcal{T})$ be a discrete hierarchy obtained by tree-sampling. Further, let $\mathcal{W}(v_i)$ be a random walk on $\mathcal{M}$ rooted in $v_i$ resulting in path $\hat{r}_i$. We have that*

$$p_{\text{anc}}^{(\mathcal{T})}(z_k | v_j) = p(z_k \in \hat{r}_i) =: p_{\text{anc}}^{(\mathcal{M})}(z_k | v_j). \tag{3}$$

*(See proof in App. A.4)*

Note that any random walk on $\mathcal{M}$ must result in a path since, because $\mathcal{M}$ is acyclic, no transient state can be visited more than one time in a random walk. Theorem 2 means that we can use ancestor probabilities arising from the tree-sampling and Markov chain views interchangeably.

### 3.2 Lowest common ancestor probabilities

For the hierarchical clustering problem we are interested in the lowest common ancestor (LCA) of two leaf nodes $v_i$ and $v_j$, which we denote by $v_i \wedge v_j$. The LCAs are required to compute both the Dasgupta cost as well as the Tree-Sampling Divergence (TSD). Therefore, when optimizing for good Dasgupta or TSD scores, it is crucial to exactly and efficiently compute the LCA probabilities that arise. We denote these LCA probabilities by $p^{(\mathcal{T})}(z_k = v_i \wedge v_j)$. Previous work resorts to heuristics in approximating the LCA probabilities (Monath et al., 2019). In contrast, our connection to absorbing Markov chains admits efficient closed-form computation of LCA probabilities, as we now show.

**Theorem 3.** *Let $A$ and $B$ be transition matrices describing a (soft or discrete) hierarchy as introduced in Section 2. Let $r_i = (r_i^{(1)}, \ldots, r_i^{(T-1)}, z_{n'})$ denote a path from leaf $v_i = r_i^{(1)}$ to the root $z_{n'}$, where $|r_i| = T$. Then, the probability of internal node $z_k$ being the lowest common ancestor $v_i \wedge v_j$ of leaf nodes $v_i \neq v_j$ under the tree-sampling procedure from Section 2.2 is*

$$p^{(\mathcal{T})}(z_k = v_i \wedge v_j) = \sum_{(r_i, r_j): z_k = v_i \wedge v_j} p(\underline{r_i}) \cdot p(\underline{r_j}), \tag{4}$$

where $\underline{r_i}$ is the part of a path from $v_i$ up to (and including) internal node $z_k$, i.e., $\underline{r_i} = (r_i^{(1)}, \ldots, z_k)$. *(See proof in App. A.5)*

For $v_i = v_j$, an LCA probability is trivially the parent probability, i.e., $p(z_k = v_i \wedge v_j) = p(z_k|v_i)$. From Eq. (4) it seems that a straightforward way to compute the LCA probabilities is to enumerate the set $\{(r_i, r_j) : z_k = v_i \wedge v_j\}$ of all pairs of potential paths from $v_i$ and $v_j$ to the internal node $z_k$ and sum their probabilities. However, this is intractable in practice because the size of the set grows exponentially in the 'depth' of the internal node $k$ (see App. A.8, Thm. 7). In Thm. 4 we show how to compute the LCA probabilities that arise when sampling pairs of random walks from $\mathcal{M}$.

**Theorem 4.** *Let $\mathcal{M}$ be a Markov chain as defined in Definition 1, $z_k$ be an internal node, and $v_i \neq v_j$ be leaf nodes. Let $\mathcal{W}(v_i), \mathcal{W}(v_j)$ be independent random walks on $\mathcal{M}$ rooted in $v_i$ and $v_j$, respectively. Then, the probability that internal node $z_k$ is the lowest internal node (as by the topological order) traversed by both random walks, i.e., the lowest common ancestor of $v_i$ and $v_j$, is*

$$p^{(\mathcal{M})}(z_k = v_i \wedge v_j) = p_{\text{anc}}(z_k|v_i)p_{\text{anc}}(z_k|v_j) - \sum_{k'=1}^{k-1} p^{(\mathcal{M})}(z_{k'} = v_i \wedge v_j)p_{\text{anc}}(z_k|z_{k'})^2 \tag{5}$$

*(See proof in App. A.6)*

Thus, due to the independence of pairs of random walks in the Markov chain $\mathcal{M}$, we can compute LCA probabilities efficiently in closed form on the Markov chain. However, the joint ancestor probability $p_{\text{anc}}^{(\mathcal{M})}(z_k|v_i, v_j)$ does not reflect the underlying process of sampling $\mathcal{T} = (\hat{A}, \hat{B}) \sim P_{A,B}(\mathcal{T})$ any more, i.e., in general $p_{\text{anc}}^{(\mathcal{M})}(z_k|v_i, v_j) = p_{\text{anc}}(z_k|v_i) \cdot p_{\text{anc}}(z_k|v_j) \neq p_{\text{anc}}^{(\mathcal{T})}(z_k|v_i, v_j)$.

As our **main theoretical result**, we show next that, remarkably, the LCA probabilities obtained from the Markov chain are indeed equivalent to the tree-sampling LCA probabilities:

**Theorem 5.** *Let $A$ and $B$ be transition matrices describing a (soft or discrete) hierarchy as described in Sec. 2. The LCA probabilities arising from tree-sampling and the Markov chain are equal, i.e.,*

$$p^{(\mathcal{T})}(z_k = v_i \wedge v_j) = p^{(\mathcal{M})}(z_k = v_i \wedge v_j). \tag{6}$$

*(See proof in App. A.7)*

This result is surprising at first, since we made the assumption that pairs of random walks are independent in the Markov chain sampling process. However, recall that two paths in the tree-sampling process are disjoint and independent *until they meet* at their LCA $z_k$. Thus, for this subset of the pairs of paths leading to $z_k$, the independence assumption in $\mathcal{M}$ does not lead to an error.

The result is also very useful in practice, since it means that we can use efficient computations from the Markov chain view to *exactly* compute the tree-sampling LCA probabilities. As we show in the following, we can jointly compute LCA probabilities for all pairs of leaves in a vectorized way.

### 3.3 Efficient Vectorized Computation

**Definition 2** (Fundamental Matrix)**.** *Let $\mathcal{M}$ be an absorbing Markov chain and $Q$ as in Definition 1. The fundamental matrix $N$ of Markov chain $\mathcal{M}$ is*

$$N \in \mathbb{R}^{|\mathcal{S}| \times |\mathcal{S}|} = (I - Q)^{-1} = \begin{bmatrix} I & A(I - B)^{-1} \\ 0 & (I - B)^{-1} \end{bmatrix}. \tag{7}$$

$N_{ij}$ equals the expected number of visits of state $j$ when starting a random walk in state $i$. Since $\mathcal{M}$ is acyclic, each transient state can be visited at most once, i.e., $N_{ij}$ is the probability of state $j$ being on a random path to the root starting from $i$. Observing the block structure of $N$ in Eq. (7), we define

$$P^{\text{anc}} \in \mathbb{R}^{n \times n'} := A(I - B)^{-1}, \qquad \tilde{P}^{\text{anc}} \in \mathbb{R}^{n' \times n'} := (I - B)^{-1} - I \tag{8}$$

$P_{ij}^{\text{anc}} = p_{\text{anc}}(z_j|v_i)$ is the probability of internal node $z_j$ ending up being an ancestor of $v_i$ under the tree-sampling procedure. Analogously, $\tilde{P}^{\text{anc}}$ provides the ancestor probabilities among the internal nodes, i.e. $\tilde{P}_{ij}^{\text{anc}} = p_{\text{anc}}(z_j|z_i)$. Note that we subtract the identity matrix in Eq. (8) since the diagonal entries of the fundamental matrix block $(I - B)^{-1}$ are always trivially 1, as they correspond to the probability of *traversing* an internal node $z_k$ when also *starting* from $z_k$. By subtracting $I$ we obtain the ancestor probabilities by enforcing that an internal node is not its own ancestor. Since $B$ is strictly upper triangular, the inverse $(I - B)^{-1}$ always exists and is efficient to compute in $O(n'^2)$.

**Theorem 6.** *The vector of LCA probabilities of all internal nodes w.r.t. leaf nodes* $\mathrm{v}_i$ *and* $\mathrm{v}_j$ *can be computed in a vectorized way via*

$$\boldsymbol{P}^{LCA}_{\mathrm{v}_i \neq \mathrm{v}_j} \in \mathbb{R}^{n'} = (\boldsymbol{P}^{anc}_{\mathrm{v}_i} \odot \boldsymbol{P}^{anc}_{\mathrm{v}_j})^T \cdot (\boldsymbol{I} + \tilde{\boldsymbol{P}}^{anc} \odot \tilde{\boldsymbol{P}}^{anc})^{-1} \qquad \boldsymbol{P}^{LCA}_{\mathrm{v}_i, \mathrm{v}_i} = \boldsymbol{A}_{\mathrm{v}i}, \qquad (9)$$

*where* $\odot$ *denotes the element-wise (Hadamard) product. (See proof in App. A.9)*

**Complexity analysis.** As we show in App. A.10, we can exploit sparsity in real-world graphs and thus do not have to construct the full $\mathbb{R}^{n \times n \times n'}$ LCA tensor $\boldsymbol{P}^{\text{LCA}}$. This leads to a complexity of $O(m \times n'^2)$ for both Soft-Das. and Soft-TSD, which is efficient since typically we have $n' \ll n$. All equations are vectorized and thus benefit from GPU acceleration. More details in App. A.12.

## 3.4 INTEGRAL SOLUTIONS

In App. A.13 we analyze the properties of the relaxed problem Soft-TSD which our method FPH optimizes. We prove in Theorem 10 that the optimization problem is *integral*, i.e., the global maximum is discrete. This is remarkable, since we are actually optimizing over *continuous* hierarchies parameterized by $\boldsymbol{A}$ and $\boldsymbol{B}$. This implies that the global maximum of the relaxed problem is the same as for the combinatorial problem of optimizing over discrete hierarchies. Soft-Das. is not obviously convex or concave, thus not obviously integral.

## 3.5 FURTHER CONSIDERATIONS

**Choice of** $n'$**.** The number of internal nodes $n'$ is an important hyperparameter of our method (as well as most baselines). Similar to, e.g., the number of clusters $k$ in $k$-means, large numbers of internal nodes $n'$ lead to more expressive hierarchies, which on the other hand are less interpretable by a human and require more memory and computation. In Fig. 2, we show how the expressiveness (as measured by TSD/Dasgupta) improves for increasing values of $n'$, and in App. B.6 we visualize learned hierarchies with different $n'$. Since FPH typically trains within a few minutes, our general recommendation is to use the elbow method (Thorndike, 1953) to determine $n'$.

**Constrained vs. unconstrained optimization.** Since our probabilistic hierarchy model leads to fully differentiable metrics (i.e., Soft-Das. and Soft-TSD), we can optimize the metrics in an end-to-end fashion via gradient descent. Note that the matrices $\boldsymbol{A}$ and $\boldsymbol{B}$ are constrained to be row-stochastic; we therefore experiment with two optimization schemes: unconstrained optimization of $\boldsymbol{A}$ and $\boldsymbol{B}$ (e.g. using Adam optimizer and softmax to obtain row-stochastic matrices), or constrained optimization via projected gradient descent (PGD). In our ablation study we found that the PGD optimization consistently leads to better results (see Fig. 4). We attribute this performance difference to the severe gradient re-scaling in the softmax operation when the parameters become very large or small, leading to very small step sizes (Niculae, 2020); thus, unless otherwise stated, FPH uses PGD optimization.

**Scaling to large graphs.** For fast training on large graphs on commodity GPUs, we propose a simple yet effective batching scheme. We uniformly pick $K$ random leaves from the graph at each iteration. Then we select the induced subgraph of these $K$ nodes and their neighbors while capping the total number of nodes by some constant $C$. We then compute the loss and perform the update based on the selected subgraph. By using our batching procedure we do not need to have all parameters on the GPU, which enables scaling to very large graphs.

**Node embeddings.** In our default setting, we directly parameterize the matrices $\boldsymbol{A}$ and $\boldsymbol{B}$ as learnable parameters via gradient descent. That is, the direct parent probabilities $p(\mathrm{z}_k|\mathrm{v}_i)$ and $p(\mathrm{z}_k|\mathrm{z}_{k'})$ are the only trainable parameters. As an alternative, we also experiment with learning a *node embedding* for each leaf and internal node. We compute the parent probabilities $\boldsymbol{A}$ and $\boldsymbol{B}$ via softmax on the negative Euclidean distances of the embeddings. Given a fixed embedding size, this leads to a parameterization that scales linearly in the number of leaf nodes and internal nodes.

**Initialization.** We have found that our model (as well as most of the baselines) can greatly benefit from a "smart" initialization scheme. For the direct parameterization, we have found initializing from the solution obtained from the average linkage algorithm (Jardine & Sibson, 1968) to work well. Unless stated otherwise, FPH uses initialization from average linkage. In contrast to vector data, linkage algorithms are fast on graph data and can be performed in $O(m)$ (Benzécri, 1982; Murtagh & Contreras, 2012), thus not affecting the complexity of FPH. In the embedding parameterization, we experiment with initializing the embeddings using DeepWalk (Perozzi et al., 2014).

| | Dasgupta cost (lower is better) | | | | | | | | Normalized TSD (higher is better) | | | | | | | |
|---|---|---|---|---|---|---|---|---|---|---|---|---|---|---|---|---|
| Alg. | Ward | Louv. | UF | HypHC | HGHC | RGHC | Avg. lk. | FPH | Ward | Louv. | UF | HypHC | HGHC | RGHC | Avg. lk. | FPH |
| Brain | 618.81 | 777.14 | 712.33 | 571.64 | 749.40 | 556.57 | 556.68 | **503.67** | 31.72 | 29.28 | 28.61 | 17.48 | 24.18 | 22.05 | 28.91 | **32.34** |
| OpenFlight | 382.45 | 633.66 | 393.58 | 463.43 | 487.96 | 488.90 | 363.40 | **355.61** | 55.48 | 51.51 | 53.89 | 39.08 | 49.50 | 39.56 | 52.02 | **57.72** |
| Genes | 202.17 | 247.26 | 251.01 | 495.26 | 366.53 | 247.07 | 196.50 | **183.63** | 66.80 | 67.47 | 62.95 | 20.66 | 53.33 | 51.81 | 66.72 | **67.69** |
| Citeseer | 92.27 | 178.23 | 98.61 | 215.62 | 150.26 | 131.89 | 83.69 | **77.16** | 69.43 | 68.45 | 67.40 | 37.22 | 57.61 | 50.61 | 67.80 | **69.37** |
| Cora-ML | 281.82 | 336.86 | 342.86 | 442.09 | 411.49 | 350.00 | 292.77 | **254.78** | 56.47 | 57.51 | 53.06 | 30.73 | 46.76 | 42.68 | 55.30 | **58.02** |
| PolBlogs | 377.63 | 443.48 | 350.74 | 330.58 | 354.86 | 433.77 | 355.61 | **262.48** | 27.54 | 25.93 | 25.23 | 22.21 | 23.94 | 19.41 | 25.25 | **31.41** |
| WikiPhysics | 736.11 | 986.32 | 753.81 | 759.07 | 840.15 | 740.87 | 658.04 | **537.95** | 45.28 | 46.03 | 43.40 | 32.02 | 39.70 | 38.39 | 43.15 | **49.97** |
| ogbn-arxiv | 22,870 | 31,655 | 52,666 | OOM | 22,076 | 24,077 | 20,671 | **14,354** | 36.77 | 37.75 | 24.75 | OOM | 26.05 | 25.21 | 33.64 | **39.66** |
| ogbl-collab | 13,835 | 20,664 | 91,807 | OOM | 34,934 | 21,057 | 15,716 | **13,493** | 45.33 | 46.12 | 27.90 | OOM | 24.80 | 34.07 | 45.44 | **48.36** |
| DBLP | 31,138 | 40,744 | 148,439 | OOM | 94,384 | 44,424 | 36,463 | **31,686** | 38.26 | 40.92 | 20.21 | OOM | 15.96 | 27.82 | 38.99 | **41.66** |

Table 1: Hierarchical clustering results ($n' = 512$). Bold/underline indicate best/second best scores.

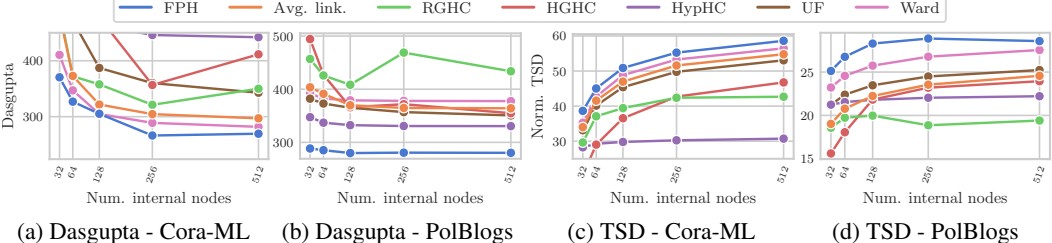

(a) Dasgupta - Cora-ML  (b) Dasgupta - PolBlogs  (c) TSD - Cora-ML  (d) TSD - PolBlogs

Figure 2: Hierarchical clustering results measured by Dasgupta cost (lower is better) and TSD (higher is better). More results in Fig. 3 in the appendix.

## 4  EXPERIMENTS

**Datasets.** We use 11 real world datasets (McCallum et al., 2000; Sen et al., 2008; Aspert et al., 2019; Amunts et al., 2013; Cho et al., 2014; Adamic & Glance, 2005; Patokallio; Wang et al., 2020; Yang & Leskovec, 2015), including the very large ogbn-products dataset with around 2.3M nodes and 62M edges (Hu et al., 2020). We always select the largest connected component (LCC) as a preprocessing step and convert each graph to an undirected one. We provide further information about the datasets in Table 6, including number of nodes, number of edges and mutual information (MI) between nodes which is an upper-bound on the TSD score (Charpentier & Bonald, 2019). In our experiments we report *normalized* TSD in percent.

**Baselines.** We compare our method with the following deep learning baselines: Routing Gradient-based clustering (**RGHC**) (Monath et al., 2017), Hyperbolic Gradient-based Clustering (**HGHC**) (Monath et al., 2019), Ultrametric Fitting (**UF**) (Chierchia & Perret, 2019), and **HypHC** (Chami et al., 2020). Importantly RGHC, HGHC and HypHC also optimize a relaxed version of the Dasgupta cost. Moreover, we compare to the average linkage algorithm (**Avg. link.**) as well as the Ward linkage algorithm (**Ward**) (Ward Jr., 1963). Finally, we compare to the Louvain method (Blondel et al., 2008) (**Louvain**). For RGHC, HGHC, and HypHC, which require vector data as input (as opposed to graphs), we use DeepWalk embeddings as node features. Note that linkage algorithms return a full hierarchy with $n' = n - 1$. Therefore we use the compression scheme introduced in (Charpentier & Bonald, 2019) to reduce the size of the hierarchies to the desired number of internal nodes. See App. B.7 for our hyperparameter choices.

For the results on Dasgupta and TSD, we train each baseline 5 times and report results for the best run.[1] For all models, we apply a post-processing treatment consisting in pruning unused internal nodes. For FPH, we select for each node its most likely parent to obtain a discrete tree.

### 4.1  RESULTS

**Hierarchical clustering.** We report results on hierarchical clustering with $n' = 512$ internal nodes in Table 1. FPH outperforms all baselines on all datasets on TSD and on all except one dataset on the Dasgupta cost. Remarkably, the average linkage algorithm outperforms most of the deep-learning based baselines. We partially attribute this to the fact that the baselines use DeepWalk embeddings as node attributes.

---

[1] Since we are not using any label information, this does not lead to data leakage or overfitting on test data; we show standard deviations in Table 11. Since FPH is deterministic, we only run it once.

| Model | Citeseer | Cora | Polblogs | ogbn-arxiv | DBLP |
|---|---|---|---|---|---|
| Avg. | 0.367 | 0.420 | 0.507 | 0.216 | 0.526 |
| RGHC | 0.281 | 0.400 | **0.730** | 0.286 | 0.510 |
| HGHC | 0.365 | 0.379 | 0.177 | 0.290 | 0.408 |
| Ward | 0.368 | 0.504 | 0.702 | **0.411** | 0.591 |
| UF | 0.347 | 0.428 | 0.676 | 0.254 | 0.598 |
| HypHC | 0.270 | 0.121 | 0.691 | OOM | OOM |
| Louvain | 0.329 | 0.500 | 0.640 | 0.395 | 0.558 |
| FPH | **0.398** | 0.462 | 0.680 | 0.251 | 0.560 |
| FPH (Louv.) | 0.380 | 0.507 | 0.614 | 0.399 | 0.564 |
| FPH (Ward) | 0.393 | **0.516** | 0.708 | 0.401 | **0.604** |

Table 2: NMI results on real-world datasets. FPH (Louv.) and FPH (Ward) refer to FPH initialized from the solutions of Louvain and Ward, respectively.

Moreover, we noticed that HypHC requires an excessive amount of triplet samples to obtain good results. Even $n^2$, as suggested by the authors, performs poorly. We use $50M$ triplets for all datasets, which is more than 4.5 times the recommended number of $n^2$ on Wiki-Physics. This took almost 24 hours to compute, which is why we could not go higher. We clearly see a dependence of HypHC's performance on the number of triplets, as it performs competitively on PolBlogs, our smallest dataset, and poorly on Wiki-Physics, the largest dataset on which we were able to run HypHC. In Tbls. 8, 9 we show baseline results with DW $d = 128$ embeddings, which did not improve results consistently.

In Figure 2 we show how the models perform for different numbers of internal nodes $n'$. As expected, the models generally obtain better scores for higher capacity. Again, we observe that FPH performs best across datasets and values of $n'$, highlighting the effectiveness of our approach. Specifically on the Dasgupta cost, FPH substantially outperforms the baselines. This is remarkable since RGHC, HGHC and HypHC also optimize a relaxed version of the Dasgupta cost.

**External evaluation.** As we typically have no knowledge about ground-truth hierarchies in real-world data, it is difficult to perform external evaluation. To address this, we propose the following:

**(i)** We compute the normalized mutual information (NMI) between the ground-truth class labels and the learned hierarchies when cutting them appropriately to divide the leaves into the same number of (flat) clusters. We set $n' = 256$ for all models (except RGHC, since it ran OOM; we use $n' = 128$ instead) and set FPH to optimize TSD. Note, though, that the node labels are not necessarily solely based on connectivity of nodes, but e.g., also on node attributes, to which the models do not have access. Thus, we cannot expect perfect correlation of "good" hierarchies in the sense of explaining edges in the graph and NMI w.r.t. ground-truth node labels. In Table 2 we show the results; the hierarchies learned by FPH achieve strong scores, highlighting its effectiveness.

**(ii)** Similarly, we can use the *ground-truth* clustering from synthetic hierarchical stochastic block-model (HSBM) (Lyzinski et al., 2016) graphs to compute the NMI scores at each level of the hierarchy. We use five HSBM graphs with 100 and 1000 nodes, respectively. The graphs have three levels (see Figs. 6,7 for example graphs). On the small graphs we set $n' = 64$; on the large graphs, we use the same settings as in **(i)**. In Table 10 we show the results. FPH outperforms all baselines and is able to recover the ground-truth hierarchies to a very high accuracy.

**Ablation study.** In our ablation study, we compare (i) FPH (i.e., with constrained optimization using PGD) vs. unconstrained optimization with Adam (FPH-U) and using softmax; (ii) average linkage initialization vs. random initialization (FPH-R, FPH-UR); (iii) direct optimization of $A$, $B$ vs. learning node embeddings (FPH Emb.). Here, we experiment with random initialization (FPH-R Emb.) as well as DeepWalk initialization (FPH Emb. DW). The results, obtained on the Wiki-Physics dataset, are displayed in Figure 4 (App.). We observe that FPH, i.e. constrained PGD optimization of $A$ and $B$ with initialization from average linkage performs best. In a similar way, "smart" initialization from DeepWalk tends to improve the results on FPH Emb., too, but the effect is less pronounced. In general, FPH variants learning embeddings perform worse than our full version; this may be due to the gradient-rescaling due to softmax, which we also suspect is a reason why unconstrained optimization achieves weaker results than FPH.

**Scalability.** We run experiments on the very large graph dataset ogbn-products, which has about 2.3M nodes and 62M edges. We train FPH both with $n'=512$ and $1024$ internal nodes performing batching as explained in Sec. 3.5. We show the results in Table 3. FPH outperforms all baselines TSD and is competitive on Dasgupta, with both $n'=512$ and $1024$. Note that we could not compare

| $n'$ | Dasgupta | | Norm. TSD | |
| | 512 | 1024 | 512 | 1024 |
| --- | --- | --- | --- | --- |
| Ward | **144,157** | **127,968** | 37.05 | 40.59 |
| HGHC | 219,959 | 168,851 | 35.90 | 39.53 |
| RGHC | 184,688 | - | 35.35 | - |
| Avg. link. | 175,571 | 168,753 | 40.33 | 43.21 |
| FPH | 147,169 | 142,404 | **43.79** | **45.57** |

Table 3: Results on ogbn-products.

| Dataset | FPH | DCSBM | DW | Ad./Ad. | VGAE |
| --- | --- | --- | --- | --- | --- |
| Cora-ML | 95.7 | 95.5 | 94.3 | 86.5 | **95.9** |
| Citeseer | **96.2** | 93.6 | 96.0 | 76.8 | 94.8 |
| PolBlogs | 94.3 | **94.9** | 84.8 | 92.6 | 92.8 |
| WikiPhysics | **97.2** | 96.9 | 92.9 | 96.6 | 97.0 |
| Brain | 94.1 | **95.2** | 83.8 | 90.7 | 93.2 |
| OpenFlight | **99.3** | 99.0 | 94.3 | 98.4 | 99.0 |
| Genes | 69.8 | 66.9 | **70.3** | 53.0 | 66.6 |

Table 4: AUC-PR score (%) for link prediction.

with HypHC, since (a) the implementation constructs a dense $n \times n$ matrix, which would require more than 14TB memory assuming single precision; (b) sampling $n^2$ triplets, as recommended by the authors, would take weeks. Further, we do not report results on UF since it did not converge. RGHC ran out of memory for $n'$=1024.

**Runtime.** Due to the efficient, vectorized computations, a full epoch of a complete dataset to compute Soft-TSD or Soft-Dasgupta using our model is very fast. A complete forward pass on Wiki-Physics with 512 internal nodes takes only about 130ms on GPU and about 2s on CPU. On the other hand, a full evaluation on ogbn-products takes only about 13 minutes (758s) on CPU, and training the model on a GPU takes approximately 3.5 hours for 2K epochs (with batching).

**Link prediction.** As the TSD score can be interpreted in terms of a reconstruction loss, we can use the scores of the reconstruction scheme in Eq. (25) (App. B.1) for predicting links. We compare with DeepWalk **(DW)** and the variational graph autoencoder **(VGAE)** (Kipf & Welling, 2016), where the link prediction score of two nodes is the cosine of their respective embeddings. Further, we compare with degree-corrected stochastic blockmodels **(DCSBM)** (Karrer & Newman, 2011) and Adamic Adar **(Ad./Ad.)** (Adamic & Adar, 2001), which are established strong baselines for link prediction in non-attributed graphs. See App. B.2 for details. Importantly, *none* of the models use node features, including FPH. As shown in Table 4, FPH achieves best or second-best performance on all datasets. This is remarkable, since FPH is not trained specifically for this task, which highlights the generality and usefulness of the hierarchies discovered by FPH.

**Qualitative analysis.** In Table 5 we provide insight into our model trained on the Cora-ML dataset. We show the first three levels of internal nodes of the learned hierarchy. Each cell corresponds to an internal node and contains the three most frequent words of the abstracts of all papers (i.e., leaf nodes) assigned to the respective internal nodes. Words from a node are excluded from its children to avoid duplicates. Terms tend from more general at the root (e.g., 'problem') towards more specific at the lower levels (e.g., 'mcmc'). Further, we can identify categories of machine learning approaches, e.g., reinforcement learning (L2, row 2), symbolic AI (L2, row 4), or

| Root | Level 1 | Level 2 |
| --- | --- | --- |
| learning, algorithm, problem | neural, paper, networks | asocs, generalization, dynamic |
| | | genetic, reinforcement, search |
| | | data, models, training |
| | | reasoning, knowledge, planning |
| | chain, markov, sampler | series, time, bayesian |
| | | gibbs, distribution, mcmc |
| | dimacs, evolutionary, species | sum, binary, comparison |
| | | perfect, graphs, parameterized |
| | | technical, report, polynomial |
| | stability, systems, linear | control, gain, output |
| | | proved, trajectory, bounded |
| | | lyapnov, state, online |
| | wavelet, minimax, estimation | |

Table 5: Cora-ML hierarchy visualization.

variational inference (L2, row 6). Again, FPH did not see any text information (node attributes) and performed the clustering based on citations (edges) alone. Besides offering qualitative evidence that the learned hierarchies are reasonable, this also showcases a potential real-world application of FPH: a scholar can explore the hierarchy of topics in an academic field to discover relevant papers.

## 5 CONCLUSION

We propose a new probabilistic model over hierarchies on graphs which can be learned using end-to-end gradient-based optimization. By drawing connections to absorbing Markov chains we can compute relevant quantities such as lowest common ancestor probabilities *exactly and efficiently* in a vectorized way. For the first time, this allows to directly optimize for relaxed versions of quality metrics for hierarchical clustering such as Dasgupta cost or Tree-Sampling Divergence (TSD) in an end-to-end fashion. Our Flexible Probabilistic Hierarchy model outperforms strong traditional as well as recent deep-learning-based baselines on nearly all datasets and tasks considered and easily scales to massive graphs with millions of nodes.

## ETHICS STATEMENT

Since our method does not directly focus on a specific real-world application, good or bad societal outcomes depend on how practitioners and researchers use it in practice. This means that it could potentially be abused, e.g., by corporations or governments, to identify groups and hierarchies of dissidents via recorded metadata. For instance, phone call metadata is routinely collected at scale by service providers and governments. By using our method on the large-scale call graph it could be possible to identify groups of political dissidents and to repress them. On the other hand, we argue that our method can also have positive impact, e.g., by making corpora of literature more accessible to users or by enabling scientists, e.g., to discover cliques and hierarchies in biological networks. Moreover, our contribution is algorithmic in nature and does not consider the effects of potential biases or discrimination in the underlying network data.

## REPRODUCIBILITY STATEMENT

For reproducibility and verifiability of our theoretical results, we provide complete proofs of all ten theorems of our work in Appendices A.3-A.13. We further make explicit all assumptions and definitions we use to derive our results. For reproducibility of our experimental results, we first highlight that our model's core implementation is a straightforward PyTorch implementation of the matrix equations in this work. Further, we detail our experimental approach in Sec. 4 and provide hyperparameter choices for our method as well as the baselines in Table 7 in the appendix. For the baselines, we use the authors' official implementations and use the suggested hyperparameters. To compute the Dasgupta and TSD metrics (as well as to obtain the results for the Louvain algorithm), we use the sknetwork Python library.[2] Our implementation is available at `https://www.daml.in.tum.de/fph`

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

## A  PROOFS AND THEORETICAL ANALYSIS

### A.1  CONTINUOUS VERSIONS OF TSD AND DASGUPTA

Our probabilistic hierarchy model enables us to replace the discrete parent assignments in the Dasgupta cost and TSD with the parent probabilities from the relaxed adjacency matrices $\boldsymbol{A}$ and $\boldsymbol{B}$. This, in turn, leads to LCA *probabilities* which are consistent under the tree-sampling procedure. For the first time, this allows us to directly and efficiently optimize for relaxed versions of hierarchical clustering quality metrics in an end-to-end fashion instead of proxy losses or heuristic algorithms.

Recall the equation of the discrete Dasgupta cost, $\mathrm{Das}(\hat{\mathcal{T}}) = \sum_{\mathrm{v}_i,\mathrm{v}_j \in V} P(\mathrm{v}_i,\mathrm{v}_j) \sum_{\mathrm{z}} \mathbb{I}_{[\mathrm{z}=\mathrm{v}_i \wedge \mathrm{v}_j]} c(\mathrm{z})$. $c(\mathrm{z})$ is the number of leaves for which internal node z is an ancestor, i.e., $c(\mathrm{z}) = \sum_{\mathrm{v} \in V} \mathbb{I}_{[\mathrm{z} \in \mathrm{anc}(\mathrm{v})]}$. Thus, we get

$$\mathrm{Das}(\hat{\mathcal{T}}) = \sum_{\mathrm{v}_i,\mathrm{v}_j \in V} P(\mathrm{v}_i,\mathrm{v}_j) \sum_{\mathrm{z}} \mathbb{I}_{[\mathrm{z}=\mathrm{v}_i \wedge \mathrm{v}_j]} \sum_{\mathrm{v} \in V} \mathbb{I}_{[\mathrm{z} \in \mathrm{anc}(\mathrm{v})]}.$$

We propose the soft Dasgupta cost **(Soft-Das)** by replacing the indicators $\mathbb{I}_{[\cdot]}$ with their expectations, i.e., with their associated probabilities:

$$\mathrm{Soft\text{-}Das}(\boldsymbol{P_{A,B}}(\mathcal{T})) = \sum_{\mathrm{v}_i,\mathrm{v}_j \in V} P(\mathrm{v}_i,\mathrm{v}_j) \sum_{\mathrm{z}} p(\mathrm{z}=\mathrm{v}_i \wedge \mathrm{v}_j) \sum_{\mathrm{v}} p_{\mathrm{anc}}(\mathrm{z}|\mathrm{v})$$

Intuitively, Soft-Das is the Dasgupta cost of the *expected hierarchy* obtained via sampling from $\boldsymbol{A}, \boldsymbol{B}$, i.e., $\mathrm{Soft\text{-}Das}(\boldsymbol{P_{A,B}}(\mathcal{T})) \equiv \mathrm{Das}(\mathbb{E}_{\hat{\mathcal{T}} \sim \boldsymbol{P_{A,B}}(\mathcal{T})}[\hat{\mathcal{T}}])$. Ideally, we would like to optimize the expected Dasgupta cost when sampling discrete hierarchies, i.e., $\mathrm{Exp\text{-}Das}(\boldsymbol{P_{A,B}}(\mathcal{T})) = \mathbb{E}_{\hat{\mathcal{T}} \sim \boldsymbol{P_{A,B}}(\mathcal{T})}\left[\mathrm{Das}(\hat{\mathcal{T}})\right]$, i.e., to put the expectation outside the Das function, which is nontrivial. We leave this extension for future work.

In the same way, we propose a differentiable version of the soft Tree Sampling Divergence **(Soft-TSD)**. To this end, we replace the discrete assignments in the distributions $p(\mathrm{z})$ and $q(\mathrm{z})$ with probabilistic assignments, i.e.

$$\mathrm{Soft\text{-}TSD}(\boldsymbol{P_{A,B}}(\mathcal{T})) = \mathrm{KL}(p(\mathrm{z})\|q(\mathrm{z})) = \sum_{\mathrm{z}} p(\mathrm{z}) \log \frac{p(\mathrm{z})}{q(\mathrm{z})}$$

$$\text{where } p(\mathrm{z}) = \sum_{\mathrm{v}_i,\mathrm{v}_j} p(\mathrm{z}=\mathrm{v}_i \wedge \mathrm{v}_j) P(\mathrm{v}_i,\mathrm{v}_j)$$

$$q(\mathrm{z}) = \sum_{\mathrm{v}_i,\mathrm{v}_j} p(\mathrm{z}=\mathrm{v}_i \wedge \mathrm{v}_j) P(\mathrm{v}_i) P(\mathrm{v}_j)$$

Analogously to above, Soft-TSD effectively computes the TSD score of the *expected* hierarchy obtained via sampling from $\boldsymbol{A}, \boldsymbol{B}$. Extending our model to compute $\mathrm{Exp\text{-}TSD}(\boldsymbol{P_{A,B}}(\mathcal{T})) = \mathbb{E}_{\hat{\mathcal{T}} \sim \boldsymbol{P_{A,B}}(\mathcal{T})}\left[\mathrm{TSD}(\hat{\mathcal{T}})\right]$ is left for future work. Note that for both Soft-TSD and Soft-Das we recover the same score as their discrete formulations in the case of a deterministic probabilistic model, i.e. when $\boldsymbol{A}$ and $\boldsymbol{B}$ are binary-valued.

### A.2  TREE-SAMPLING PROCEDURE

Recall our assumption that the internal nodes are ordered, and that $B_{ij} = 0$ if $j \leq i$. This implies that there are no possible cycles, or equivalently that $\boldsymbol{B}$ is a strictly upper-triangular matrix, i.e., it describes a directed acyclic graph (DAG). Combined with the fact that each node in a tree (except the root) has *exactly* one parent, we see that the sampled discrete hierarchy is indeed a tree. We denote this tree-sampling process by $\mathcal{T} = (\hat{\boldsymbol{A}}, \hat{\boldsymbol{B}}) \sim P_{\boldsymbol{A,B}}(\mathcal{T})$. We can also compute the probability of any tree $\mathcal{T}$ under the sampling procedure described above:

$$P_{\boldsymbol{A,B}}(\mathcal{T} = (\hat{\boldsymbol{A}}, \hat{\boldsymbol{B}})) = \prod_{i,j} A_{i,j}^{\hat{A}_{i,j}} \prod_{i',j'} B_{i',j'}^{\hat{B}_{i',j'}} \tag{10}$$

Note that $A_{i,j}$ and $B_{i',j'}$ are the probabilities of internal nodes $z_j$ and $z_{j'}$ to be a parent of leaf and internal nodes $v_i$ and $z_{i'}$ respectively while $\hat{A}_{i,j}$ and $\hat{B}_{i',j'}$ are equal to 1 if these connections exist in the tree $\mathcal{T}$, else 0.

### A.3 PROOF OF THEOREM 1

*Proof.* For $\mathcal{M}$ to be absorbing (i) it must have at least one absorbing state, and (ii) at least one absorbing state must be reachable from any state in a finite number of steps. For (i), $\omega$ is an absorbing state since its self-transition probability $T_{k,k} = 1$, where $k = |\mathcal{S}| + 1$ is its corresponding index in the transition matrix $T$. Thus, once reached, a random walk cannot leave the state $\omega$. To show (ii), we note that since the transient state transition matrix $Q$ is a strictly upper-triangular matrix, any random walk on $\mathcal{M}$ must lead to state $z_{n'}$. From state $z_{n'}$, the random walk transits to $\omega$ with probability $w_{n'} = 1$. Thus, since $n'$ is finite, state $\omega$ can be reached from any state in a finite number of steps. Further, $\omega$ is the only absorbing state since all self-transition probabilities of states in $\mathcal{S}$ are zero as $\text{diag}(Q) = 0$. Since $Q$ is strictly upper-triangular, none of the transient states can be visited more than once on a random walk, and therefore $\mathcal{M}$ is acyclic. □

### A.4 PROOF OF THEOREM 2

*Proof.* We can arbitrarily define the order in which we sample from the categorical distributions in $A$ and $B$ because of the independence of the sampling steps. We choose to start by sampling first from $A_i$, i.e., the row corresponding to the leaf node $v_i$ under consideration: $w^{(1)} \sim \text{Cat}(A_i)$. Next, we sample from the row corresponding to $w^{(1)}$, and repeat until we reach $z_k$, i.e.,

$$w^{(t)} \sim \text{Cat}(B_{w^{(t-1)}}) \quad \text{for } 1 < t \leq T,$$

where $w^{(T)} = z_k$. For the remaining entries, we continue in arbitrary order. Observe that $(w^{(1)}, \ldots, w^{(T)})$ are the ancestors of leaf $v_i$ in $\mathcal{T}$. Further observe that this sampling procedure is identical to how the path $\hat{r}_i$ is generated in the random walk $\mathcal{W}(v_i)$, completing the proof. □

### A.5 PROOF OF THEOREM 3

*Proof.* First, recall that all paths end in the root node, such that $r_i$ necessarily ends in $z_{n'} = r_i^{(T)}$. When reasoning about the lowest *common* ancestors, it is no longer sufficient to consider the ancestors (or, equivalently, path to the root) of a single leaf node in isolation. Instead, we need to consider *pairs* of dependent paths $(r_i, r_j), i \neq j$ rooted in $v_i$ and $v_j$, respectively. Note that $r_i$ and $r_j$ necessarily converge at some internal node $z_k = v_i \wedge v_j$ — the latest at the root node $z_{n'}$.

Thus, we denote with $\underline{r_i} = (r_i^{(1)}, \ldots, v_i \wedge v_j)$ the first part of the path $r_i$ until (and including) its lowest common ancestor with $r_j$, i.e., $v_i \wedge v_j$. Analogously, $\overline{r_i} = (r_i^{(|\underline{r_i}|+1)}, \ldots, z_{n'})$, such that $r_i = (\underline{r_i}, \overline{r_i})$. Further, note that the paths $r_i$ and $r_j$ are on the same underlying hierarchy $\mathcal{T}$, thus $\overline{r_i} = \overline{r_j}$, as both paths have the same trajectory to the root once they have reached their lowest common ancestor, i.e., they are dependent.

The probability of observing the pair of paths $(r_i, r_j)$ under the tree-sampling perspective is

$$p^{(\mathcal{T})}((r_i, r_j)) = p(r_i^{(1)}|v_i) \cdot \prod_{t=2}^{|\underline{r_i}|} p(r_i^{(t)}|r_i^{(t-1)}) \cdot p(r_j^{(1)}|v_j) \cdot \prod_{t=2}^{|r_j|} p(r_j^{(t)}|r_j^{(t-1)}) \cdot \prod_{t=|\underline{r_i}|+1}^{|r_i|} p(r_i^{(t)}|r_i^{(t-1)}) \tag{11}$$

More compactly,

$$p^{(\mathcal{T})}((r_i, r_j)) = p(\underline{r_i}) \cdot p(\underline{r_j}) \cdot p(\overline{r_i}) = p((\underline{r_i}, \underline{r_j})) \cdot p(\overline{r_i}) = p((\underline{r_i}, \underline{r_j})) \cdot p(\overline{r_j}) \tag{12}$$

Importantly, we can see from Eq. (12) that, in general, $p^{(\mathcal{T})}((r_i, r_j)) \neq p(r_i) \cdot p(r_j)$ i.e., the paths $r_i$ and $r_j$ are *not* independent. We denote $p_{\text{anc}}^{(\mathcal{T})}(z_k|v_i, v_j)$ the probability of the internal node $z_k$ to be the ancestor of leaf nodes $v_i$ and $v_j$ under the tree-sampling perspective. Hence, the probability of the internal node $z_k$ to be the ancestor of leaf nodes $v_i$ and $v_j$ under dependent and independent random walks are different i.e. $p_{\text{anc}}^{(\mathcal{T})}(z_k|v_i, v_j) \neq p_{\text{anc}}(z_k|v_i) \cdot p_{\text{anc}}(z_k|v_j)$. This makes intuitive sense

because knowing that $z_{k'}$ is an ancestor of $v_i$ and $v_j$ in a tree $\mathcal{T}$, additional knowledge that $z_k, k > k'$ is an ancestor of $v_i$ implies that $z_k$ is also an ancestor of $v_j$.

However, for the parts of $r_i$ and $r_j$ *before* they converge, Eq. (12) shows that $p((\underline{r_i}, \underline{r_j})) = p(\underline{r_i}) \cdot p(\underline{r_j})$. This is because all transitions in $\underline{r_i}$ and $\underline{r_j}$ are disjoint thus independent. This is an important insight because it means that the probability of observing two paths both converging at an internal node $z_k$ factorizes. $\square$

## A.6 PROOF OF THEOREM 4

*Proof.* We start by reorganizing Eq. (5):

$$\overbrace{p_{\text{anc}}(z_k|v_i) p_{\text{anc}}(z_k|v_j)}^{(i)} = \overbrace{p^{(\mathcal{M})}(z_k = v_i \wedge v_j)}^{(ii)} + \overbrace{\sum_{k'=1}^{k-1} p^{(\mathcal{M})}(z_{k'} = v_i \wedge v_j) p_{\text{anc}}(z_k|z_{k'})^2}^{(iii)} \tag{13}$$

In words, we can split the event "$z_k$ is an ancestor of $v_i$ and $v_j$" (i) into two mutually exclusive events: (ii) $z_k$ is the lowest common ancestor of $v_i$ and $v_j$; or (iii) some internal node lower in the topological order is the LCA of $v_i$ and $v_j$, and further, both random walks also traverse through $z_k$. (ii) and (iii) are mutually exclusive since exactly one internal node is the LCA for $v_i$ and $v_j$ on any two random walks.

Since the two random walks are independent, the probability of $z_k$ being traversed on both walks factorizes. Thus, $p_{\text{anc}}^{(\mathcal{M})}(z_k|v_i, v_j) = p_{\text{anc}}(z_k|v_i) p_{\text{anc}}(z_k|v_j)$ and therefore (i) is the probability of $z_k$ being an ancestor of $v_i$ and $v_j$ in our Markov chain $\mathcal{M}$.

The events in (iii) can indeed be expressed:

$$p^{(\mathcal{M})}\left(z_{k'} = v_i \wedge v_j, z_k \in \text{anc}\,(v_i, v_j)\right) = p^{(\mathcal{M})}\left(z_{k'} = v_i \wedge v_j\right) \cdot p^{(\mathcal{M})}\left(z_k|z_{k'} \in \text{anc}\,(v_i, v_j)\right) \tag{14}$$

where in the last step we exploit that $z_{k'} = v_i \wedge v_j$ implies $z_{k'} \in \text{anc}\,(v_i, v_j)$ as well as the Markov property of the random walks. Further, note that

$$\begin{aligned} p_{\text{anc}}^{(\mathcal{M})}(z_k|z_{k'} \in \text{anc}(v_i, v_j)) &= p_{\text{anc}}(z_k|z_{k'} \in \text{anc}(v_i)) \cdot p_{\text{anc}}(z_k|z_{k'} \in \text{anc}(v_j)) \\ &= p_{\text{anc}}(z_k|z_{k'})^2, \end{aligned} \tag{15}$$

where we first exploit factorization due to independence and in the last step again the Markov property of the random walks. $\square$

## A.7 PROOF OF THEOREM 5

*Proof.* Let $\hat{r}_i \in \mathcal{P}(v_i)$, $\hat{r}_j \in \mathcal{P}(v_j)$ be two independent random walks on $\mathcal{M}$ rooted in $v_i$ and $v_j$, respectively. Then,

$$p^{(\mathcal{M})}(z_k = v_i \wedge v_j) = \sum_{(\hat{r}_i, \hat{r}_j): z_k = v_i \wedge v_j} p((\underline{\hat{r}_i}, \underline{\hat{r}_j})), \tag{16}$$

where $\underline{\hat{r}_i} = (\hat{r}_i^{(1)}, \dots, z_k)$ is the first part of $\hat{r}_i$ until it reaches $z_k$. Note that the second part of the paths, $\overline{\hat{r}_i}$ and $\overline{\hat{r}_j}$, which are theoretically independent under our Markov chain model, are marginalized out in the LCA formula.

However, due to the independence of the first part of the paths $\underline{\hat{r}_i}$ and $\underline{\hat{r}_j}$ under both models (see Eq. (4)), we can write:

$$\begin{aligned} p^{(\mathcal{M})}(z_k = v_i \wedge v_j) &= \sum_{(\hat{r}_i, \hat{r}_j): z_k = v_i \wedge v_j} p((\underline{\hat{r}_i}, \underline{\hat{r}_j})) \\ &= \sum_{(\hat{r}_i, \hat{r}_j): z_k = v_i \wedge v_j} p(\underline{\hat{r}_i}) \cdot p(\underline{\hat{r}_j}) \\ &= p^{(\mathcal{T})}(z_k = v_i \wedge v_j). \quad \square \end{aligned} \tag{17}$$

## A.8 Number of pairs of LCA paths.

**Theorem 7.** *Let $\mathcal{M}$ be a Markov chain as defined in Definition 1. The number of pairs of paths from two leaves $v_i$, $v_j$ for which an internal node $z_k$ is the lowest common ancestor is $3^{k-1}$.*

*Proof.* Proof by induction over $k$. For the base case $k = 1$ we have one pair of paths for which $z_k$ is the LCA, i.e. directly from $v_i$ to $z_k$ and $v_j$ to $z_k$. Assume that for internal node $z_k$ there are $3^{k-1}$ unique pairs of paths for which $z_k$ is the LCA. For each of these paths we can generate three unique paths for which $z_{k+1}$ is the LCA. (1) rewire the last transition of $v_i$'s path to go to $z_{k+1}$ instead of $z_k$. (2) do the same but for $v_j$. (3) rewire both $v_i$'s and $v_j$'s last transition to go to $z_{k+1}$ instead of $z_k$. Thus, the number of pairs of paths from $v_i$ and $v_j$ to $z_{k+1}$ is $3 \cdot 3^{k-1} = 3^k$. $\square$

## A.9 Proof of Theorem 6

We provide here the proof for the fast vectorized computation of $\boldsymbol{P}^{\text{LCA}}_{v_i,v_j}$ in Theorem. 6.

*Proof.* For the case $v_i \neq v_j$, we start by reorganizing Eq. (9):

$$\boldsymbol{P}^{\text{anc}}_{v_i} \odot \boldsymbol{P}^{\text{anc}}_{v_j} = \boldsymbol{P}^{\text{LCA}}_{v_i,v_j} + \boldsymbol{P}^{\text{LCA},T}_{v_i,v_j} \cdot \tilde{\boldsymbol{P}}^{\text{anc}} \odot \tilde{\boldsymbol{P}}^{\text{anc}} \tag{18}$$

Note that the inverse $(\boldsymbol{I} + \tilde{\boldsymbol{P}}^{\text{anc}} \odot \tilde{\boldsymbol{P}}^{\text{anc}})^{-1}$ is guaranteed to exist and is efficient to compute because $\tilde{\boldsymbol{P}}^{\text{anc}} \odot \tilde{\boldsymbol{P}}^{\text{anc}}$ is a strictly upper triangular and therefore nilpotent matrix. The $k$-th entry is thus:

$$\left[\boldsymbol{P}^{\text{anc}}_{v_i} \odot \boldsymbol{P}^{\text{anc}}_{v_j}\right]_k = \boldsymbol{P}^{\text{LCA}}_{v_i,v_j,v_k} + \sum_{k'=1}^{n'} \boldsymbol{P}^{\text{LCA}}_{v_i,v_j,v_{k'}} \cdot \left[\tilde{\boldsymbol{P}}^{\text{anc}} \odot \tilde{\boldsymbol{P}}^{\text{anc}}\right]_{k',k}.$$

Plugging in the definitions of Eq. (8), using Theorem 2, and observing that due to the upper triangular structure $\tilde{\boldsymbol{P}}^{\text{anc}}_{k',k} = 0$ for $k' > k$ we obtain

$$p_{\text{anc}}(z_k|v_i) \cdot p_{\text{anc}}(z_k|v_j) = p(z_k = v_i \wedge v_j) + \sum_{k'=1}^{k-1} p(z_{k'} = v_i \wedge v_j) \cdot p_{\text{anc}}(z_k|z_{k'})^2.$$

For $v_i = v_j$, we have $\boldsymbol{P}^{\text{LCA}}_{v_i,v_i} = \boldsymbol{A}_{v_i}$, i.e., again simply the parent probabilities of $v_i$. $\square$

## A.10 Vectorized computations

All quantities involved in Soft-Das and Soft-TSD can be computed in closed-form based on the Markov chain $\mathcal{M}$ and its fundamental matrix. However, their computation should not be done naively, as this involves unnecessary computations. Constructing the full tensor of LCA probabilities is expensive since $\boldsymbol{P}^{\text{LCA}} \in \mathbb{R}^{n \times n \times n'}$. Note, however, that to compute the Soft Dasgupta loss or the distribution $p(z)$ in TSD we only require the LCA probabilities $p(z = v_i \wedge v_j)$ for pairs of leaves connected by an edge (i.e., $P(v_i, v_j) > 0$). That is, we only need to construct an LCA probability matrix of shape $\mathbb{R}^{m \times n'}$. Thus, we can exploit the sparsity of real world graphs, as typically $m \ll n^2$.

In a similar way, the computation of the distribution $q(z)$ does also not require the expensive explicit computation of $p(z_k = v_i \wedge v_j)$ for all pairs of leaf nodes. Instead, we can again exploit insights from the Markov chain $\mathcal{M}$. First, observe that the equation of $q(z)$ in Soft-TSD describes an expectation:

$$q(z) = \mathbb{E}_{v_i,v_j \sim P(v)} \left[p(z = v_i \wedge v_j)\right]. \tag{19}$$

Defining $\hat{\boldsymbol{p}}^{\text{anc}} = \boldsymbol{p}^T \cdot \boldsymbol{P}^{\text{anc}}$, the computation of this expectation can be vectorized similarly to Theorem 6 (see derivation in App. A.11):

$$\boldsymbol{q} = ((\hat{\boldsymbol{p}}^{\text{anc}} \odot \hat{\boldsymbol{p}}^{\text{anc}})^T - (\boldsymbol{p} \odot \boldsymbol{p})^T \cdot \boldsymbol{P}^{\text{anc}} \odot \boldsymbol{P}^{\text{anc}}) \cdot (\boldsymbol{I} + \tilde{\boldsymbol{P}}^{\text{anc}} \odot \tilde{\boldsymbol{P}}^{\text{anc}})^{-1} + (\boldsymbol{p} \odot \boldsymbol{p})^T \boldsymbol{A}.$$

### A.11 Vectorized $q$ computation.

We provide here the proof for the fast vectorized computation of $q$ in Eq. 19.

*Proof.* We first rewrite the expectation Eq. 19 in vectorized form:

$$q = \sum_{i,j} \boldsymbol{p}_{\mathrm{v}_i} \boldsymbol{P}^{\mathrm{LCA}}_{\mathrm{v}_i, \mathrm{v}_j} \boldsymbol{p}_{\mathrm{v}_j}$$

where we denote $P(\mathrm{v}_i) = \boldsymbol{p}_{\mathrm{v}_i}$. Subsequently, we can plug the $\boldsymbol{P}^{\mathrm{LCA}}$ formula Eq. (9) and pull $\boldsymbol{p}_{\mathrm{v}_i}$ into the Hadamard product which is done over the internal node dimension.

$$\hat{q} = \sum_{i,j} \boldsymbol{p}_{\mathrm{v}_i} \boldsymbol{p}_{\mathrm{v}_j} (\boldsymbol{P}^{\mathrm{anc}}_{\mathrm{v}_i} \odot \boldsymbol{P}^{\mathrm{anc}}_{\mathrm{v}_j})^T \cdot (\boldsymbol{I} + \tilde{\boldsymbol{P}}^{\mathrm{anc}} \odot \tilde{\boldsymbol{P}}^{\mathrm{anc}})^{-1}$$

$$= \sum_i \boldsymbol{p}_{\mathrm{v}_i} \left( \boldsymbol{P}^{\mathrm{anc}}_{\mathrm{v}_i} \odot \sum_j \boldsymbol{p}_{\mathrm{v}_j} \boldsymbol{P}^{\mathrm{anc}}_{\mathrm{v}_j} \right)^T \cdot (\boldsymbol{I} + \tilde{\boldsymbol{P}}^{\mathrm{anc}} \odot \tilde{\boldsymbol{P}}^{\mathrm{anc}})^{-1}$$

$$= \left( \sum_i \boldsymbol{p}_{\mathrm{v}_i} \boldsymbol{P}^{\mathrm{anc}}_{\mathrm{v}_i} \odot \sum_j \boldsymbol{p}_{\mathrm{v}_j} \boldsymbol{P}^{\mathrm{anc}}_{\mathrm{v}_j} \right)^T \cdot (\boldsymbol{I} + \tilde{\boldsymbol{P}}^{\mathrm{anc}} \odot \tilde{\boldsymbol{P}}^{\mathrm{anc}})^{-1}$$

$$= (\hat{\boldsymbol{p}}^{\mathrm{anc}} \odot \hat{\boldsymbol{p}}^{\mathrm{anc}})^T \cdot (\boldsymbol{I} + \tilde{\boldsymbol{P}}^{\mathrm{anc}} \odot \tilde{\boldsymbol{P}}^{\mathrm{anc}})^{-1}.$$

where $\hat{\boldsymbol{p}}^{\mathrm{anc}} = \boldsymbol{p}^T \cdot \boldsymbol{P}^{\mathrm{anc}}$. However, recall that for $\mathrm{v}_i = \mathrm{v}_j$ the LCA probabilities are $\boldsymbol{A}_{\mathrm{v}_i}$; thus, we need to correct for this difference to obtain $q$:

$$\hat{q} = \sum_{i \neq j} \boldsymbol{p}_{\mathrm{v}_i} \boldsymbol{P}^{\mathrm{LCA}}_{\mathrm{v}_i, \mathrm{v}_j} \boldsymbol{p}_{\mathrm{v}_j} + \sum_i \boldsymbol{p}_{\mathrm{v}_i} \hat{\boldsymbol{P}}^{\mathrm{LCA}}_{\mathrm{v}_i, \mathrm{v}_i} \boldsymbol{p}_{\mathrm{v}_i}$$

$$= q - \sum_i \boldsymbol{p}_{\mathrm{v}_i} \boldsymbol{P}^{\mathrm{LCA}}_{\mathrm{v}_i, \mathrm{v}_i} \boldsymbol{p}_{\mathrm{v}_i} + \sum_i \boldsymbol{p}_{\mathrm{v}_i} \hat{\boldsymbol{P}}^{\mathrm{LCA}}_{\mathrm{v}_i, \mathrm{v}_i} \boldsymbol{p}_{\mathrm{v}_i}$$

$$\Leftrightarrow q = \hat{q} + \sum_i \boldsymbol{p}_{\mathrm{v}_i} \boldsymbol{P}^{\mathrm{LCA}}_{\mathrm{v}_i, \mathrm{v}_i} \boldsymbol{p}_{\mathrm{v}_i} - \sum_i \boldsymbol{p}_{\mathrm{v}_i} \hat{\boldsymbol{P}}^{\mathrm{LCA}}_{\mathrm{v}_i, \mathrm{v}_i} \boldsymbol{p}_{\mathrm{v}_i} = \hat{q} + \sum_i \boldsymbol{p}_{\mathrm{v}_i} \boldsymbol{A}_{\mathrm{v}_i} \boldsymbol{p}_{\mathrm{v}_i} - \sum_i \boldsymbol{p}_{\mathrm{v}_i} \hat{\boldsymbol{P}}^{\mathrm{LCA}}_{\mathrm{v}_i, \mathrm{v}_i} \boldsymbol{p}_{\mathrm{v}_i}$$

$$= \hat{q} + (\boldsymbol{p} \odot \boldsymbol{p})^T \cdot \boldsymbol{A} - (\boldsymbol{p} \odot \boldsymbol{p})^T \hat{\boldsymbol{P}}^{\mathrm{LCA}}_{\mathrm{v}_i, \mathrm{v}_i}$$

$$= ((\hat{\boldsymbol{p}}^{\mathrm{anc}} \odot \hat{\boldsymbol{p}}^{\mathrm{anc}})^T - (\boldsymbol{p} \odot \boldsymbol{p})^T \cdot \boldsymbol{P}^{\mathrm{anc}} \odot \boldsymbol{P}^{\mathrm{anc}}) \cdot (\boldsymbol{I} + \tilde{\boldsymbol{P}}^{\mathrm{anc}} \odot \tilde{\boldsymbol{P}}^{\mathrm{anc}})^{-1} + (\boldsymbol{p} \odot \boldsymbol{p})^T \boldsymbol{A},$$

where $\hat{\boldsymbol{P}}^{\mathrm{LCA}}_{\mathrm{v}_i, \mathrm{v}_i} = \boldsymbol{P}^{\mathrm{anc}} \odot \boldsymbol{P}^{\mathrm{anc}} \cdot (\boldsymbol{I} + \tilde{\boldsymbol{P}}^{\mathrm{anc}} \odot \tilde{\boldsymbol{P}}^{\mathrm{anc}})^{-1}$ (obtained by using Eq. equation 9). $\quad\square$

Note that, unless stated otherwise, we use $\hat{q}$ as a proxy for $q$ in our experiments for simplicity.

### A.12 Complexity analysis.

Both Dasgupta and TSD computations require to compute the ancestor probabilities (Eq. 8) which can be done in $O(n \times n'^2)$ (i.e. inverse of triangular matrix $(\boldsymbol{I} - \boldsymbol{B}) \in \mathbb{R}^{n' \times n'}$, plus matrix multiplication with $\boldsymbol{A} \in \mathbb{R}^{n \times n'}$). Then, Soft-Das. or the distribution $p(\mathrm{z})$ for the Soft-TSD loss require the LCA probabilities (Eq. 9) for all leaves connected by an edge only, amounting to $O(m \times n'^2)$ operations, where $m$ is the number of edges in the graph. Note that similarly to Eq. (8), the inverse computation in Eq. (9) can be done in $O(n'^2)$. Additionally, Soft-TSD requires the computation of $q(\mathrm{z})$ ( complexity $O(n \times n'^2)$). Both Soft-Das. and Soft-TSD computations are dominated by the $O(m \times n'^2)$ term. This leads to an efficient time complexity as long as we assume a small number of internal nodes $n' \ll n$, which is reasonable in practice.

### A.13 Convexity of Soft-TSD

**Theorem 8.** *Let $\boldsymbol{H} \in [0,1]^{n' \times n \times n}$ be a tensor whose elements $\boldsymbol{H}_{kij} = p(\mathrm{z}_k = \mathrm{v}_i \wedge \mathrm{v}_j)$ are the LCA probabilities of internal nodes w.r.t. pairs of leaf nodes. Soft-TSD$(\boldsymbol{H})$ is convex in $\boldsymbol{H}$.*

*Proof.* Let $\boldsymbol{H}^{(1)}$, $\boldsymbol{H}^{(2)}$ be two LCA probability tensors defined as above, and $0 \leq \alpha \leq 1$. We first compute the distribution $p$ induced by the edge distribution. We have that

$$
\begin{aligned}
p(\alpha \boldsymbol{H}_k^{(1)} + (1-\alpha)\boldsymbol{H}_k^{(2)}) &= \sum_{\mathrm{v}_i, \mathrm{v}_j} P(\mathrm{v}_i, \mathrm{v}_j)(\alpha \boldsymbol{H}_{kij}^{(1)} + (1-\alpha)\boldsymbol{H}_{kij}^{(2)}) \\
&= \alpha \sum_{\mathrm{v}_i, \mathrm{v}_j} P(\mathrm{v}_i, \mathrm{v}_j)\boldsymbol{H}_{kij}^{(1)} + (1-\alpha) \sum_{\mathrm{v}_i, \mathrm{v}_j} P(\mathrm{v}_i, \mathrm{v}_j)\boldsymbol{H}_{kij}^{(2)} \quad (20) \\
&= \alpha \cdot p(\boldsymbol{H}_k^{(1)}) + (1-\alpha) \cdot p(\boldsymbol{H}_k^{(2)}).
\end{aligned}
$$

Analogously we compute the distribution $q$ induced by the independent node distribution. We have that

$$
q(\alpha \boldsymbol{H}_k^{(1)} + (1-\alpha)\boldsymbol{H}_k^{(2)}) = \alpha \cdot q(\boldsymbol{H}_k^{(1)}) + (1-\alpha) \cdot q(\boldsymbol{H}_k^{(2)}). \quad (21)
$$

We combine this with the well-known fact that KL-divergence is convex w.r.t. pairs of distributions, i.e.,

$$
\mathrm{KL}(\alpha p_1(z) + (1-\alpha)p_2(z), \alpha q_1(z) + (1-\alpha)q_2(z)) \leq \alpha \mathrm{KL}(p_1(z), q_1(z)) + (1-\alpha)\mathrm{KL}(p_2(y), q_2(z)),
$$

to obtain the desired result:

$$
\text{Soft-TSD}(\alpha \boldsymbol{H}^{(1)} + (1-\alpha)\boldsymbol{H}^{(2)}) \leq \alpha \cdot \text{Soft-TSD}(\boldsymbol{H}^{(1)}) + (1-\alpha) \cdot \text{Soft-TSD}(\boldsymbol{H}^{(2)}). \quad (22)
$$

$\square$

**Theorem 9.** *Let $\mathcal{H}_{\mathrm{FPH}} = \{\boldsymbol{H} : \exists \boldsymbol{A}, \boldsymbol{B} \in \Phi(n, n') : \boldsymbol{H} = \mathrm{FPH}(\boldsymbol{A}, \boldsymbol{B})\}$ denote the set of probabilistic hierarchies which can be represented by FPH. Here, $\Phi(n, n')$ are the constraints FPH places on $\boldsymbol{A}, \boldsymbol{B}$ (see Sec. 2), and FPH $(\boldsymbol{A}, \boldsymbol{B})$ is shorthand the mapping from transition matrices to lowest common ancestor probability tensors defined in Theorems 4 and 6. Here, $\boldsymbol{H}_{kij} = p(\mathrm{z}_k = \mathrm{v}_i \wedge \mathrm{v}_j)$ are the LCA probabilities of internal nodes w.r.t. pairs of leaf nodes. This set $\mathcal{H}_{\mathrm{FPH}}$ is convex.*

*Proof.* We start by recalling form Theorems 4 and 5 that FPH computes lowest common ancestor probabilities $p(\mathrm{z}_k = \mathrm{v}_i \wedge \mathrm{v}_j)$ from the continuous parent probability matrices $\boldsymbol{A}$ and $\boldsymbol{B}$ such that the LCA probabilities are consistent with the expected result from the tree-sampling procedure described in Sec. A.2. More formally,

$$
\begin{aligned}
\boldsymbol{H}_{kij} := p(\mathrm{z}_k = \mathrm{v}_i \wedge \mathrm{v}_j) &= \mathbb{E}_{\hat{\boldsymbol{H}} \sim (\boldsymbol{A}, \boldsymbol{B})} \left[ \mathbb{I}[\mathrm{z}_k = \mathrm{v}_i \wedge \mathrm{v}_j] \right] \\
&= \mathbb{E}_{\hat{\boldsymbol{H}} \sim (\boldsymbol{A}, \boldsymbol{B})} \left[ \hat{\boldsymbol{H}}_{kij} \right] = \mathbb{E} \left[ \hat{\boldsymbol{H}} \right]_{kij},
\end{aligned} \quad (23)
$$

where $\hat{\boldsymbol{H}} \in \{0, 1\}^{n' \times n \times n}$ is a discrete hierarchy obtained via tree-sampling from $\boldsymbol{A}$ and $\boldsymbol{B}$. By definition of the expectation we write

$$
\boldsymbol{H} = \mathbb{E}_{\hat{\boldsymbol{H}} \sim (\boldsymbol{A}, \boldsymbol{B})} \left[ \hat{\boldsymbol{H}} \right] = \sum_{\hat{\boldsymbol{H}} \in \mathcal{H}(n, n')} p(\hat{\boldsymbol{H}} | \boldsymbol{A}, \boldsymbol{B}) \cdot \hat{\boldsymbol{H}}, \quad (24)
$$

where $\mathcal{H}(n, n')$ is the set of all valid discrete hierarchies with $n$ leafs and $n'$ internal nodes. Thus, any continuous hierarchy $\boldsymbol{H}$ learned by FPH is a convex combination of discrete hierarchies $\hat{\boldsymbol{H}}$. This completes the proof. $\square$

**Theorem 10.** *The Soft-TSD optimization problem solved by FPH is integral. That is, the global maximum of the Soft-TSD optimization problem solved by FPH is the same as the global optimum of the* discrete *optimization problem of optimizing TSD over discrete hierarchies.*

*Proof.* This follows from Theorems 8 and 9. Theorem 8 establishes that the Soft-TSD objective function is convex in the hierarchy tensors $\boldsymbol{H}$; Theorem 9 proves that the set of hierarchies FPH optimizes over is convex. When maximizing a convex function over a convex set, we are guaranteed to find the global optimum at a vertex of the constraint set, which are discrete hierarchies in the case of FPH. Thus, the global maximizer is a discrete hierarchy; this discrete hierarchy must also be the maximizer of the discrete TSD optimization problem, since our relaxation optimizes over a superset of all discrete hierarchies. $\square$

| Dataset | Nodes (LCC) | Edges (LCC) | MI (LCC) | License |
|---|---|---|---|---|
| *PolBlogs*, (Adamic & Glance, 2005) | 1,222 | 16,715 | 2.39 | n/a |
| *Brain*, Amunts et al. (2013) | 1,770 | 8,957 | 3.37 | n/a |
| *Citeseer*, Sen et al. (2008) | 2,110 | 3,694 | 5.69 | n/a |
| *Genes*, Cho et al. (2014) | 2,194 | 2,688 | 6.12 | n/a |
| *Cora-ML*, McCallum et al. (2000); Bojchevski & Günnemann (2018) | 2,810 | 7,981 | 5.23 | n/a |
| *WikiPhysics*, Aspert et al. (2019) | 3,309 | 31,251 | 3.44 | n/a |
| *OpenFlight*, Patokallio, | 3,097 | 18,193 | 3.44 | ODbL |
| *Ogbn-products*, Hu et al. (2020) | 2,385,902 | 61,806,367 | 9.29 | Amazon license |
| *Ogbn-arxiv*, Hu et al. (2020); Wang et al. (2020) | 169,343 | 1,157,799 | 7,40 | ODC-BY |
| *Ogbl-collab*, Hu et al. (2020); Wang et al. (2020) | 232,865 | 961,883 | 9.02 | ODC-BY |
| *DBLP*, Yang & Leskovec (2015) | 317,080 | 1,049,866 | 9.64 | n/a |

Table 6: Dataset summary; we convert directed datasets to undirected and select the largest connected component (LCC).

**Implications of Theorems 8, 9, and 10.** In the previous theorems, we have shown that we are *maximizing* a convex function over a convex set. In general, maximizing a convex function over a convex set is NP-hard (Benson, 1995). Thus, we cannot hope to efficiently recover the global optimum. However, our continuous relaxation brings several practical benefits for the optimization.

First, observe that directly optimizing over the convex set of continuous hierarchies described in Theorem 9 is not practical. This is because there are exponentially many corners of the set, and encoding the constraints of the set is very difficult. Our parameterization of (continuous) hierarchies via $A$, $B$ and being able to efficiently compute the *expected* lowest common ancestor probabilty tensor enables us to optimize over a fairly low-dimensional and convex set. The constraints on $A, B$, i.e., entries in $[0, 1]$, unit row sums and upper-triangular structure of $B$, are easy to encode and enforce during optimization. This comes at the cost that mapping from $A$ and $B$ to the LCA tensor $H$ is nonconvex (yet describes, as per Theorem 9, a convex set over hierarchies). Thus, we can solve the optimization problem with off-the-shelf methods such as projected gradient descent and benefit from the elaborate techniques from nonconvex optimization. While it is possible that FPH gets stuck in a non-discrete local optima during optimization, we can easily obtain a discrete and valid hierarchy given the non-discrete local optimizer via tree-sampling or selecting the most likely parent for all leaves and internal nodes under $A$ and $B$, as described in Sec. 2.2.

## B  EXPERIMENT INFORMATION

### B.1  LINK PREDICTION WITH SOFT-TSD

The TSD can be interpreted in terms of retrieved information when reconstructing the original graph from the tree representation (Charpentier & Bonald, 2019). In this case, the reconstruction scheme for the edge weights of the reconstructed graph $\hat{\mathcal{G}}$ is:

$$\hat{w}(\mathbf{v}_i, \mathbf{v}_j) = w(\mathbf{v}_i)w(\mathbf{v}_j)\frac{p(\mathbf{v}_i \wedge \mathbf{v}_j)}{q(\mathbf{v}_i \wedge \mathbf{v}_j)} \tag{25}$$

### B.2  LINK PREDICTION SETUP

For all datasets, we randomly select 10% of edges to hold out for testing while making sure that the graph remains connected. Further, we set $n' = 256$ and minimize Soft-TSD via FPH. For DC-SBM, we use the Python package 'graph-tool' and follow the documentation[3] with default parameters to learn the model. For VGAE, we use the default hyperparameters by the authors (one hidden layer, latent dimensions $[32, 16]$, learning rate 0.01, training for 200 epochs). We use the variant described in the paper which replaces the node attributes by the $n \times n$ identity matrix. For DeepWalk, we set the embedding dimension to 10.

### B.3  DATASET SUMMARY

See Table 6 for an overview of the datasets we used.

---

[3]https://graph-tool.skewed.de/static/doc/demos/inference/inference.html

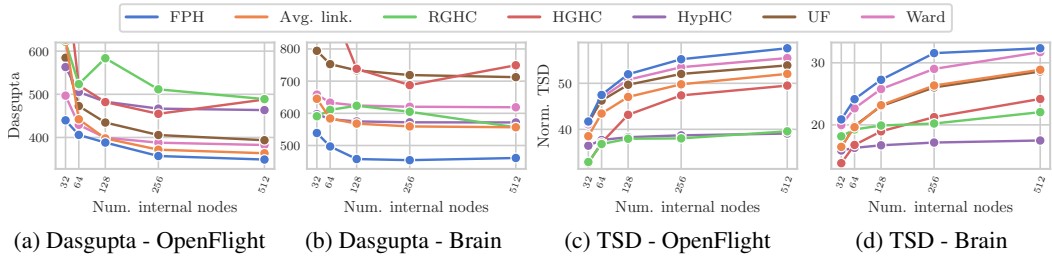

(a) Dasgupta - OpenFlight    (b) Dasgupta - Brain    (c) TSD - OpenFlight    (d) TSD - Brain

Figure 3: Results on hierarchical clustering measured by Dasgupta cost (lower is better) and TSD (higher is better).

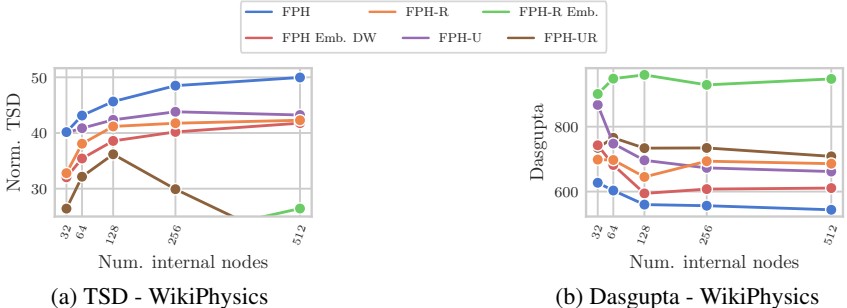

(a) TSD - WikiPhysics    (b) Dasgupta - WikiPhysics

Figure 4: Ablation study results.

## B.4 ADDITIONAL RESULT FIGURES

In Fig. 3 we show results for four more datasets.

## B.5 ABLATION STUDY

See Fig 4 for the comparison of different FPH model variants, and our full discussion in Sec. 4.1.

## B.6 HIERARCHY VISUALIZATION

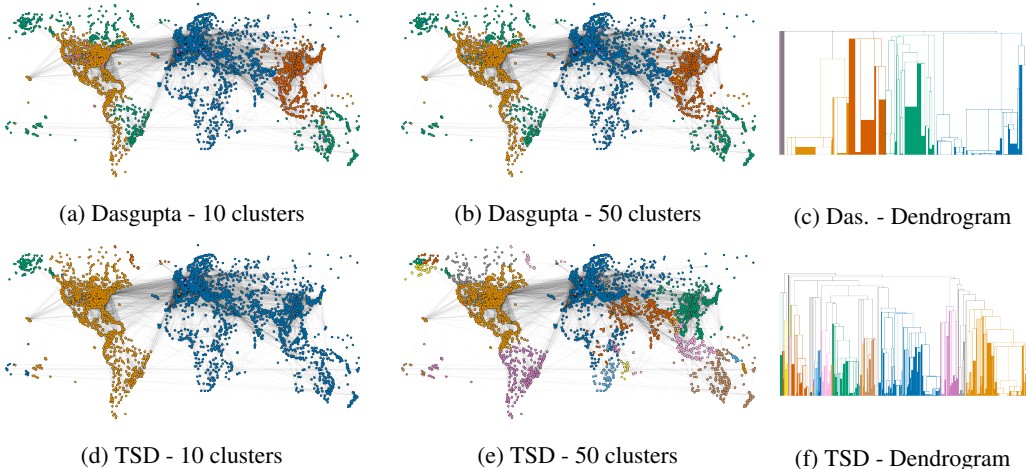

(a) Dasgupta - 10 clusters    (b) Dasgupta - 50 clusters    (c) Das. - Dendrogram

(d) TSD - 10 clusters    (e) TSD - 50 clusters    (f) TSD - Dendrogram

Figure 5: Visual comparison of trees obtained after Soft-TSD and Soft-Dasgupta optimization on OpenFlight.

We conduct a qualitative study of the structure discovered by FPH. In Figure 5 we compare the hierarchies learned by FPH when optimizing for TSD or Dasgupta, respectively. We cut at different levels of the dendrogram to obtain a coarse hierarchy (10 clusters) and fine-grained structure (50

clusters). Comparing Figure 5 (a) and (d), we notice that the coarse structure learned by optimizing Soft-Dasgupta looks more appealing, as TSD essentially splits only into the Americas and the rest of the world. At 50 clusters, however, we observe the opposite: TSD splits the airports across the world into meaningful, coherent geographical regions, whereas Dasgupta looks mostly unchanged from the coarse version, highlighting the complementarity of both quality metrics. In addition, the dendrogram learned by TSD in (f) appears to be of higher quality and more balanced than the Dasgupta dendrogram in (c).

## B.7 HYPERPARAMETERS

We use the hyperparameters for models and baselines described in Tab. 7. For smaller datasets, we train FPH for 1,000 epochs and restore the best hierarchy after training. For ogbn-products, ogbn-arxiv, ogbl-collab, and DBLP, we train for 2,000 epochs. Similarly, we use different learning rates

| Model | Hyperparameter | Value |
|---|---|---|
| FPH (TSD) | Learning rate | 150 |
| | Batch size $K$** | 10,000 |
| | Batch cutoff $C$** | 200,000 |
| FPH-R (TSD) | Learning rate | 200 |
| FPH (Das.) | Learning rate | 0.05 |
| | Batch size $K$** | 10,000 |
| | Batch cutoff $C$** | 200,000 |
| FPH Emb. | Learning rate | 0.1 |
| RGHC | Routing NN dim | 128 |
| | Iterations | 5000 |
| | Learning rate | 0.0001 |
| HGHC | Init. method | K-means + agglom. linkage |
| | Iterations | 10 |
| | Learning rate | 0.1 |
| UF | Loss | Closest + cluster size |
| | Epochs | 500 |
| | Learning rate | 0.1 |
| HypHC | Num. triples | 50M |
| | Epochs | 50 |
| | Learning rate | 0.001 |
| | Temperature | 0.1 |
| DeepWalk | Embedding dim | 10 |
| | Embedding dim* | 32 |

\* Used for ogbn-products, ogbn-arxiv, ogbl-collab, DBLP.
\** Used for ogbn-products

Table 7: Hyperparameter settings.

for $A$ and $B$ for FPH (Das.) on ogbn-arxiv, ogbl-collab, ogbn-products, and DBLP ($\mathrm{lr}_A = 1e - 2$, $\mathrm{lr}_B = 1e - 9$).

## B.8 COMPUTING INFRASTRUCTURE

We train all models on a single GPU (NVIDIA GTX 1080 Ti or NVIDIA GTX 2080 Ti, 11 GB memory) in our own in-house compute cluster. The machines have 10-core Intel CPUs. We use Python 3 and PyTorch for all our experiments.

## B.9 HSBM GRAPHS

We generated HSBMs with $n = 100$ leaf nodes and $n = 1000$ leaf nodes for our external evaluation. The small HSBMs have 3 levels with edge probabilities in $[.01, .1, .3, .6]$, a branching factor of 2 and core community sizes in $[10, 15]$. The large HSBMs have 3 levels with edge probabilities in $[.001, .01, .1, .4]$, branching factor in $[2, 3, 4]$ and core community sizes in $[30, 35]$. In Fig. 6 and

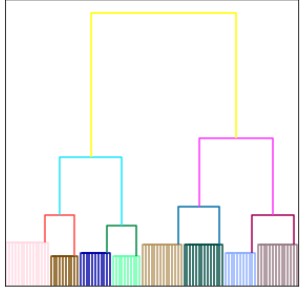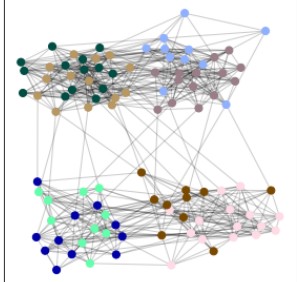

Figure 6: Example HSBM graph with $n = 100$, $n' = 7$, and three levels in the hierarchy.

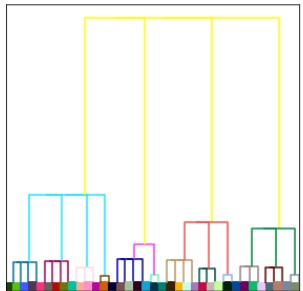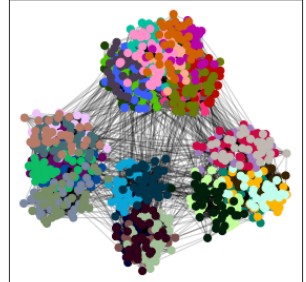

Figure 7: Example HSBM graph with $n = 1000$, $n' = 53$, and three levels in the hierarchy.

Fig. 7, we plot one of the five synthetic HSBM graphs used in our experiments (right), and their corresponding dendrograms (left). The graphs have $n = 100$ and $n = 1000$ leaf nodes and three levels of hierarchy.

## C    ADDITIONAL RESULTS

| Alg. | Dasgupta cost (lower is better) | | | | | Normalized TSD (higher is better) | | | | |
|---|---|---|---|---|---|---|---|---|---|---|
| | Ward | UF | HypHC | HGHC | RGHC | Ward | UF | HypHC | HGHC | RGHC |
| Brain | 596.73 | 938.49 | 568.18 | 894.87 | 650.53 | 32.43 | 26.28 | 17.68 | 17.31 | 16.54 |
| OpenFlight | 416.05 | 643.45 | 423.80 | 477.51 | 469.98 | 55.59 | 49.88 | 40.06 | 47.52 | 45.84 |
| Genes | 221.76 | 258.21 | 467.12 | 482.11 | 444.82 | 66.87 | 63.73 | 23.94 | 50.59 | 40.98 |
| Citeseer | 105.12 | 280.66 | 271.80 | 224.63 | 200.91 | 69.28 | 62.95 | 31.74 | 52.53 | 47.66 |
| Cora-ML | 301.47 | 673.27 | 441.21 | 516.87 | 499.49 | 57.22 | 47.96 | 29.10 | 42.10 | 35.19 |
| PolBlogs | 383.51 | 726.34 | 334.69 | 428.52 | 376.94 | 27.01 | 10.73 | 21.60 | 20.30 | 20.78 |
| WikiPhysics | 808.87 | 958.20 | 701.14 | 919.27 | 790.39 | 45.54 | 41.55 | 33.85 | 34.51 | 36.51 |
| ogbn-arxiv | 22,046 | 64,950 | OOM | 37,177 | 26,286 | 37.43 | 26.22 | OOM | 17.55 | 25.20 |
| ogbl-collab | 14,834 | 101,562 | OOM | 112,048 | 17,964 | 45.20 | 30.50 | OOM | 11.11 | 37.73 |
| DBLP | 33,349 | 160,742 | OOM | 171,975 | 41,796 | 38.87 | 22.62 | OOM | 5.61 | 29.9 |

Table 8: Hierarchical clustering results ($n' = 512$, $d = 128$).

| Model | Citeseer | Cora | Polblogs | DBLP | ogbn-arxiv |
|---|---|---|---|---|---|
| RGHC | 0.218 | 0.394 | 0.756 | 0.510 | 0.358 |
| HGHC | 0.304 | 0.362 | 0.604 | 0.655 | 0.385 |
| Ward | 0.363 | 0.445 | 0.436 | 0.587 | 0.402 |
| UF | 0.180 | 0.242 | 0.102 | 0.395 | 0.143 |
| HypHC | 0.285 | 0.390 | 0.740 | - | - |

Table 9: NMI results for $d = 128$ DeepWalk embeddings.

| | $n = 100$ | | | $n = 1000$ | | |
|---|---|---|---|---|---|---|
| Model | Level 1 | Level 2 | Level 3 | Level 1 | Level 2 | Level 3 |
| Avg. | 0.985 | 0.952 | 0.791 | 0.990 | 0.978 | 0.965 |
| RGHC | 1.0 | 0.881 | 0.766 | 0.567 | 0.897 | 0.745 |
| HGHC | 0.874 | 0.872 | 0.705 | 0.840 | 0.942 | 0.786 |
| Ward | 1.0 | 0.975 | 0.765 | 1.0 | 0.991 | 0.841 |
| UF | 1.0 | 0.921 | 0.707 | 1.0 | 0.987 | 0.791 |
| HypHC | 1.0 | 0.892 | 0.768 | 0.933 | 0.846 | 0.702 |
| Louvain | 0.693 | 0.955 | 0.795 | 0.678 | **1.0** | 0.993 |
| FPH | 1.0 | **0.994** | **0.829** | 1.0 | **1.0** | **0.994** |

Table 10: NMI results on synthetic HSBM graphs.

| | TSD standard deviation | | | Dasgupta standard deviation | | |
|---|---|---|---|---|---|---|
| Alg. | HypHC | HGHC | RGHC | HypHC | HGHC | RGHC |
| Brain | 0.53 (3%) | 0.06 (<0.5%) | 0.6 (3%) | 19.96 (3%) | 38.56 (5%) | 14.18 (3%) |
| OpenFlight | 1.15 (3%) | 0.4 (1%) | 0.7 (2%) | 26.61 (6%) | 31.14 (6%) | 23.9 (5%) |
| Genes | 0.61 (3%) | 0.12 (<0.5%) | 1.19 (2%) | 13.3 (3%) | 2.88 (1%) | 17.31 (7%) |
| Citeseer | 0.19 (1%) | 0.09 (<0.5%) | 1.29 (3%) | 6.93 (3%) | 5.56 (4%) | 8.02 (6%) |
| Cora-ML | 0.96 (3%) | 0.13 (<0.5%) | 0.61 (1%) | 17.39 (4%) | 21.78 (5%) | 12.98 (4%) |
| PolBlogs | 0.13 (1%) | 0.09 (<0.5%) | 0.18 (1%) | 4.04 (1%) | 1.92 (1%) | 3.4 (1%) |
| WikiPhysics | 0.47 (1%) | 0.14 (<0.5%) | 0.56 (1%) | 5.77 (1%) | 6.83 (1%) | 30.88 (4%) |
| ogbn-arxiv | - | 0.15 (1%) | 1.31 (5%) | - | 405 (2%) | 765 (3%) |
| ogbl-collab | - | 0.16 (1%) | 0.61 (2%) | - | 1,592 (5%) | 625 (3%) |
| DBLP | - | 0.47 (3%) | 0.81 (3%) | - | 2,482 (3%) | 1,005 (2%) |

Table 11: Standard deviations of non-deterministic baselines. In parentheses we report the standard deviation in relation to the best value reported in Table 1 in percent.

