# OpenReview forum: "End-to-End Learning of Probabilistic Hierarchies on Graphs"
_ICLR.cc/2022/Conference — ICLR 2022 Poster_

### Official Review · Reviewer_1sXn · 2021-11-01

**Correctness:** 3
**Technical Novelty And Significance:** 3
**Empirical Novelty And Significance:** 3
**Recommendation:** 6
**Confidence:** 4

**Details Of Ethics Concerns:**

Despite that the paper is mostly algorithmic and mathematical, the papers took the time to point out some instances of possible negative use of the model. The paper highlights the positive impact of the work.

**Main Review:**

Among the strengths of this paper I can mention that:
Using the adjacency information of the graph structure to identify tree-based hierarchies can be useful and computationally a good strategy as proved in other works.
The use of a Markov chain over the tree defines an acyclic chain that could be used for inference.
The block formulation of the vectorized computation could be useful for efficient implementations.
With respect to the weaknesses, there are a few items that could be clarified. Namely:
The paper states that in early experiments “PGD optimization leads to faster and better convergence” can you ellaborate what the early experiments entailed?
The paper states that the baselines were trained 5 times and results were reported for the best run, could you also report the variance for the baseline and the proposed model to have an estimate of the stability of the results?
Why were only a few datasets selected to report in certain experiments? e.g. in Table 2 only three datasets appear.

**Summary Of The Paper:**

This paper presents a hierarchical clustering over graphs using a probabilistic model via continuous relaxation of the discrete tree-based hierarchies. It uses two metrics the Dasgupta cost and the Tree-Sampling Divergence. The paper presents an efficient vectorized computation and discusses several details on the choice of hyper parameters. The article presents empirical evaluation on 8 real datasets and compares the technique with several baselines on hierarchical clustering (using NMI), comparing also results on controlled clusters (ground truth class labels, synthetic stochastic block-model) and link prediction (best or second based performance), as well as on meta-analysis which includes ablation, scalability, runtime, and a qualitative assessment.

**Summary Of The Review:**

The paper has some strengths that could be useful in applications of graph or network clustering and inference. I find the continuous tree-based hierarchy and associated Markov chain interesting and results promising. I do have some reservations about some decisions made about what to report as I highlighted above. Overall, it may be an interesting addition to ICLR if the authors can improve on the weaknesses I pointed out.

---

> ### Author Response · Authors · 2021-11-23
> **Authors' response**
>
> Thank you for pointing out this sentence where our phrasing is imprecise. What we refer to here are our ablation study results (see Fig. 4 in the appendix), which we used to decide whether to use constrained optimization with PGD optimization or unconstrained parameterization and optimization via Adam. In these experiments we find that the unconstrained (i.e., softmax) parameterization (FPH-U and FPH-UR) consistently perform worse than FPH and FPH-R, their constrained counterparts which are optimized with PGD. We have updated the description as well as provide a pointer to Fig. 4 in Section 3.5, paragraph "Constrained vs. unconstrained optimization" in the revised manuscript.
>
> > The paper states that the baselines were trained 5 times and results were reported for the best run, could you also report the variance for the baseline and the proposed model to have an estimate of the stability of the results?
>
> In the following Table we show the standard deviations of 5 runs for non-deterministic baselines. We have also added this table to the appendix (Table 13) of the revised manuscript.
>
> | Alg.|HypHC|HGHC|RGHC||HypHC|HGHC|RGHC|
> |-|-|-|-|-|-|-|-|
> |Brain|0.53 (3%)|0.06 (0%)|0.6 (3%)||19.96 (3%)|38.56 (5%)|14.18 (3%)|
> |OpenFlight|1.15 (3%)|0.4 (1%)|0.7 (2%)||26.61 (6%)|31.14 (6%)|23.9 (5%)|
> |Genes|0.61 (3%)|0.12 (<0.5%)|1.19 (2%)||13.3 (3%)|2.88 (1%)|17.31 (7%)|
> |Citeseer|0.19 (1%)|0.09 (<0.5%)|1.29 (3%)||6.93 (3%)|5.56 (4%)|8.02 (6%)|
> |Cora-ML|0.96 (3%)|0.13 (<0.5%)|0.61 (1%)||17.39 (4%)|21.78 (5%)|12.98 (4%)|
> |PolBlogs|0.13 (1%)|0.09 (<0.5%)|0.18 (1%)||4.04 (1%)|1.92 (1%)|3.4 (1%)|
> |WikiPhysics|0.47 (1%)|0.14 (<0.5%)|0.56 (1%)||5.77 (1%)|6.83 (1%)|30.88 (4%)|
> |ogbn-arxiv| OOM | 0.15 (1%)| 1.31 (5%)| | OOM| 405 (2%) | 765 (3%) |
> |ogbl-collab| OOM | 0.16 (1%)| 0.61 (2%)| | OOM| 1,592 (5%) | 625 (3%) |
> |DBLP| OOM | 0.47 (3%)| 0.81 (3%)| | OOM| 2,482 (3%) | 1,005 (2%) |
>
> **Table A**: Standard deviations of hierarchical clustering results. In parentheses we relate the standard deviations to the values reported in Table 1 of the paper. Columns 2-4 show the standard deviations of TSD and the last three columns those of Dasgupta.
>
> We see that the baselines' results are quite stable and all standard deviations are smaller than 10% of the values in Table 1, most even below 5%. We would like to emphasize again that **FPH's optimization is deterministic** as we perform full-batch gradient descent and initialize from a deterministic solution (average linkage), which is why we do not report standard deviations for FPH. We added a pointer to these results to Section 4 of the revised manuscript.
>
>
>
>
> > Why were only a few datasets selected to report in certain experiments? e.g. in Table 2 only three datasets appear.
>
> In Table 2 we show only the datasets from Table 1 where we have node labels available, which are Cora-ML, PolBlogs, and Citeseer. Among the datasets we newly added based on Reviewer knNt's suggestions, ogbn-arxiv and DBLP also come with node labels. We therefore also added the NMI results for these two new datasets to the revised manuscript (Table 12 in the appendix). For convenience, we display them here:
>
> |Model|ogbn-arxiv|DBLP|
> |-|-|-|
> |Avg.|0.216|0.526|
> |RGHC|0.286|0.510|
> |HGHC|0.290|0.408|
> |Ward|**0.411**|0.591|
> |UF|0.254|*0.598*|
> |HypHC|OOM|OOM|
> |Louvain|0.395|0.558|
> |FPH|0.251|0.560|
> |FPH (Louv.)|0.399|0.564|
> |FPH (Ward)|*0.401*|**0.604**|
>
> **Table B**: NMI results on two new datasets with node labels. **Bold** / *italic* indicate best / second best scores. FPH (Louv.) and FPH (Ward) are our model initialized from Louvain or Ward, respectively.
>
> We see that FPH can improve the performance even of strong baselines when used as initialization, which none of the baselines can easily do. With this, FPH scores best or second-best on both new datasets.
>
> **Additional experiments**. In addition, we kindly refer you the general response to all reviewers where we present additional results (suggested by one of the reviewers) which highlight that **(i)** FPH outperforms the diverse and strong baselines also on three additional datasets; **(ii)** FPH outperforms the additional Louvain baseline in all cases on hierarchical clustering; **(iii)** increasing DeepWalk dimension to $d=128$ does not improve results consistently.
>
> If you have any remaining or follow-up questions, we are happy to provide further information.

---

> > ### Comment · Reviewer_1sXn · 2021-11-23
> > **Illustrative**
> >
> > Thank you for taking the time to create your comprehensive answers. I recommend incorporating these new tables and insights, preferable, to the main body of the paper. If that is accomplish, I recommend acceptance of this paper.

---

> > > ### Author Response · Authors · 2021-11-24
> > > **Authors' response**
> > >
> > > Thank you very much for your swift reply. We are glad you found our response illustrative and comprehensive. **Note that all of the additional results** are shown in the revised version of the paper: we added results for the **three new datasets to Table 1**, and added results of the **Louvain baseline to Tables 2 and 3**. The remainder is summarized in **Tables 9-13 in the appendix**, and we have added pointers to them in the main text.
> > >
> > > Going forward, we are **eager to add** the contents of Table 9 (Louvain TSD + Dasgupta) to the main results in Table 1 in the main text (and potentially the standard deviations of Table 13, if space allows), and the NMI results of the new datasets (now in Table 12) to Table 2 in the main text.
> > > Unfortunately, **OpenReview does not allow us to update the paper** at this point and until after acceptance. We hope you can trust our (public) commitment to make the changes stated above for the camera ready version.
> > >
> > > Does this adequately address your request and lead you to recommend acceptance (i.e., raise your score to "accept")? If not, please let us know if there is anything else you would like to see.

---

### Official Review · Reviewer_8G7Q · 2021-11-03

**Correctness:** 4
**Technical Novelty And Significance:** 4
**Empirical Novelty And Significance:** Not applicable
**Recommendation:** 8
**Confidence:** 3

**Main Review:**

1) This paper is well-written and organized. The notations used throughout the paper is consistent and easy to follow. 2) Instead of providing a heuristic driven method, this method proposed in this paper has solid theoretical supports. 3) The model optimizes directly for the objectives in an end-to-end fashion. It is scalable and achieves state-of-art performance compared with both past and current baselines. 4) The experimentation setting is complete and very informative.

However, the implementation part could add more details. Also, the implementation/code is claimed to be present in the supplement materials but it is not.

**Summary Of The Paper:**

The paper proposes a probabilistic model of tree-based hierarchical clustering by considering a continuous relaxation. The authors establishes the theorical connections between tree-sampling and the Markov chain that is specifically constructed. This allows an end-to-end gradient-based learning to optimize the internal metrics of graph clustering, namely the Dasgupta cost and Tree Sampling Divergence. This method allows efficient and effective modeling of the graph clustering. Extensive experiments results on different types of graph, including some of the large scale ones, suggests the proposed method behave favorably.

**Summary Of The Review:**

This paper proposes a novel graph clustering method with solid theoretical supports and strong practical performance. The paper itself provides complete reasonings of the motivations and derivations. Overall, it is a impressive paper that brings huge improvement to an important problem of graphs.

---

> ### Author Response · Authors · 2021-11-23
> **Authors' response**
>
> Thank you for your review. We are happy you find our paper well-written and that you appreciate the theoretical support of our method.
>
> We understood that you have questions about some implementation details of our method. Regarding our **code**, in our paper we state that we share the implementation *privately* with the reviewers on OpenReview. What we meant by that is that we add **a private comment here in the forum**. We are sorry for the confusion. We made the comment on Oct 12 -- **please let us know in case you are not able to see the comment or access our code**. We clarify this sentence in the revised manuscript and further added some more details to the Reproducibility statement. We are happy to answer any additional questions you may have about our implementation.
>
> **Additional experiments.**
> In addition, we kindly refer you the general response to all reviewers where we present additional results (suggested by one of the reviewers) which highlight that **(i)** FPH outperforms the diverse and strong baselines also on three additional datasets; **(ii)** FPH outperforms the additional Louvain baseline in all cases on hierarchical clustering; **(iii)** increasing DeepWalk dimension to $d=128$ does not improve results consistently.
>
> If you have any remaining or follow-up questions, we are happy to provide further information.

---

### Official Review · Reviewer_knNt · 2021-11-12

**Correctness:** 3
**Technical Novelty And Significance:** 3
**Empirical Novelty And Significance:** 3
**Recommendation:** 6
**Confidence:** 4

**Main Review:**

The idea of hierarchical probabilistic clustering inference from the trees seems novel and interesting. Dasgupta loss is a relatively new development in the clustering literature; the contributions of this work are timely.
Note: since this is an emergence review, I did not check correctness of the proofs in the Appendix.

The experimental section is very unconvincing, however. The comparisons are performed only wrt. primitive graph clustering methods; for instance, in a graph clustering paper I would expect to find heuristic greedy modularity optimization (Blondel et al., 2008), hierarchical SBM fitting, and possibly some non-hierarchical algorithms.

Crucially, the metric space clustering baselines are given very suboptimal features for the clustering: DeepWalk's d=10 is far too low. The original DW paper uses d=128 for all graph sizes, and neural embedding-based models are known to perform poorly in low dimensional regime.

The scalability of the method is concerning as well. While the paper claims scalability, it is only being evaluated on one large-scale dataset, which is not supporting the claim.

Based on two findings above, I am ready to raise my score, given that the paper is updated with:
- A re-run of the experiments, or at least a sample of them (Table 2), with d=128, as is default for DeepWalk.
- A comparison (Tables 2-3, and possibly 5) some classical graph baselines for clustering, including Louvain.
- Results from OBG-arXiv to Table 2, and add at least one-two more medium-scale dataset (# of nodes 50k+) without graph subsampling tricks.

In the current state, it is very hard to find support for the claims in the paper experimentally. I am sorry for asking quite a lot in a short timeframe; I hope the authors/ACs do understand my reasoning. In case authors think my asks are unreasonable, please suggest some alternative ways to prove the points that are currently weak.

**Summary Of The Paper:**

The paper proposes a probabilistic clustering method based on two losses: Dasgupta loss and Tree-sampling divergence. A scalable sampling-based optimization is proposed. The modes is optimized with the projected gradient descent.

**Summary Of The Review:**

The idea and the model in the paper sound reasonable; however, the experimental results are very unconvincing: (1) Three of the baselines are operating on the embedding of the data potentially built by an ill-configured model (2) the claimed scalability is only empirically supported on *one* dataset of size 100000 (3) there is a ton of missed baselines from the graph clustering literature.

After seeing the rebuttal I raise my score to the accept category.

---

> ### Author Response · Authors · 2021-11-23
> **Authors' response (1/2)**
>
> Thank you for your constructive and helpful comments. We are happy you find our approach novel, interesting, and timely. Your concerns appear to be mostly about the experimental evaluation. We gladly received your suggestions and conducted a number of additional experiments, which, as we believe, further highlight the effectiveness and scalability of our approach.
> ### DeepWalk dimension
> Based on the reviewer's suggestion, we re-ran experiments where baselines use DeepWalk embeddings with dimension $d=128$. The results on TSD, Dasgupta, and NMI are in the appendix of the revised manuscript; for convenience, we also display them here:
>
> ||Ward|UF|HypHC|HGHC|RGHC||Ward|UF|HypHC|HGHC|RGHC|
> |-|-|-|-|-|-|-|-|-|-|-|-|
> |Brain|**596.73**|938.49|**568.18**|894.87|650.53||**32.43**|26.28|**17.68**|17.31|16.54|
> |OpenFlight|416.05|643.45|**423.80**|**477.51**|**469.98**||**55.59**|49.88|**40.06**|47.52|**45.84**|
> |Genes|221.76|258.21|**467.12**|482.11|444.82||**66.87**|**63.73**|**23.94**|50.59|40.98|
> |Citeseer|105.12|280.66|271.80|224.63|**200.91**||69.28|62.95|31.74|52.53|47.66|
> |Cora-ML|301.47|673.27|**441.21**|516.87|499.49||**57.22**|**47.9**6|29.10|42.10|35.19|
> |PolBlogs|383.51|726.34|334.69|428.52|**376.94**||27.01|10.73|21.60|20.30|**20.78**|
> |WikiPhysics|808.87|958.20|**701.14**|919.27|790.39||**45.54**|41.55|**33.85**|34.51|36.51|
> |ogbn-arxiv|**22,046**|64,950|OOM|37,177|26,286||**37.43**|**26.22**|OOM|17.55|25.20|
> |ogbl-collab|14,834|101,562|OOM|112,048|**17,964**||45.20|**30.50**|OOM|11.11|**37.73**|
> |DBLP|33,349|160,742|OOM|171,975|**41,796**||**38.87**|**22.62**|OOM|5.61|**29.9**|
>
> **Table A**: Hierarchical clustering results for baselines using DW and $d=128$. **Bold numbers** indicate improved performance for $d=128$
>
> |Model|Citeseer|Cora|Polblogs|DBLP|ogbn-arxiv|
> |-|-|-|-|-|-|
> |RGHC|0.218|0.394|**0.756**|0.510|**0.358**|
> |HGHC|0.304|0.362|**0.604**|0.655|0.385|
> |Ward|0.363|0.445|0.436|0.587|0.402|
> |UF|0.180|0.242|0.102|0.143|**0.395**|
> |HypHC|0.285|0.390|**0.740**|OOM|OOM|
>
> **Table B**: NMI results for $d=128$.
>
> Note that these results include three additional datasets (see "Scalability / new datasets"). Increasing the embedding dimension did not consistently improve the results for any of the respective baselines; for some baselines (UF, HGHC), results are worse for almost all datasets. In total, we observe an improvement in 38/119 cases, i.e. about one third. We hypothesize that this might be because some of the baselines suffer from the curse of dimensionality, where the Euclidean distances (between the DeepWalk embeddings) tend to become more similar for higher dimensions. Importantly, **none of the baselines outperform FPH on Dasgupta or TSD** in the $d=128$ scenario. We have added a pointer to these tables to Section 4.1 of the revised manuscript.
>
> ### Additional baselines
> Based on the reviewer's suggestion, we additionaly conducted experiments with the Louvain algorithm. Again, these results are in the appendix (Table 9) of the revised manuscript and displayed here for convenience. In parentheses we show for comparison our model's results and bold the best score. Note here we include results on three additional datasets (see "Scalability / new datasets" for info).
>
> |Dataset|Dasgupta; lower is better|Norm. TSD; larger is better|
> |-|-|-|
> |CoraML|336.86 (**269.50**)|57.51 (**58.62**)|
> |Citeseer|178.23 (**83.59** )|68.45 (**69.95**)|
> |Polblogs|443.48 (**295.44**)|25.93 (**28.56**)|
> |Brain|777.14 (**461.17**)|29.28 (**32.64**)|
> |Genes|247.26 (**184.35**)|67.47 (**67.65**)|
> |WikiPhysics|986.32 (**543.95**)|46.03 (**50.00**)|
> |Openflight|633.66 (**348.78**)|51.51 (**57.56**)|
> |ogbn-arxiv|31,655 (**14,796**)|37.75 (**38.79**)|
> |ogbl-collab|20,664 (**13,661**)|46.12 (**48.02**)|
> |DBLP|40,744 (**32,137**)|40.92 (**41.33**)|
>
> **Table C**: Louvain hierarchical clustering results. In parentheses we show FPH's results and we bold the best scores.
>
> The Louvain baseline scores quite competitively overall but our method outperforms it on all of the datasets and clustering metrics. Still, we believe that these results are a valuable addition to the paper, and if the reviewers agree we will move these results into Table 1 in the main text for the camera-ready version.

---

> > ### Author Response · Authors · 2021-11-23
> > **Authors' response (2/2)**
> >
> > We also compare against Louvain on NMI:
> >
> > |Dataset|NMI; higher is better|
> > |-|-|
> > |Citeseer|0.329  (**0.398**)|
> > |Cora|0.500  (0.462 / **0.507**)|
> > |Polblogs|0.640  (**0.680**)|
> > |ogbn-arxiv|0.395  (0.251 / **0.399**)|
> > |DBLP|0.558    (**0.561**)|
> > |HSBM (`n=100`, L3)|0.795  (**0.829**)|
> > |HSBM (`n=1000`, L3)|0.993  (**0.994**)|
> >
> > **Table D**: Louvain NMI results.
> >
> > In parentheses we show again FPH's results. Note again the additional datasets (ogbn-arxiv and DBLP; ogbl-collab does not have node labels) Here, our method outperforms Louvain in 5/7 cases, again highlighting the quality of FPH's learned hierarchies. On ogbn-arxiv, our method suffers from a poor initialization by average linkage (0.216). The second number (0.399) shows our method's results when we initialize from Louvain's solution. Similarly, FPH improves Louvain's solution on Cora-ML. This shows that our method can improve results of very strong baselines to further increase the quality of the learned hierarchies, which none of the baselines can do. We added Louvain's results to Tables 2 and 3 of the revised manuscript.
> >
> >
> >
> > ### Scalability / new datasets
> > We additionaly ran experiments for our method as well as all baselines on three additional datasets:
> > * `ogbn-arxiv` ($N_{LCC}=169,343$, $E_{LCC}=1,157,799$),
> > * `ogbl-collab` ($N_{LCC}=232,865$, $E_{LCC}=961,883$), and
> > * `DBLP` [[Yang & Leskovec, 2015]](https://snap.stanford.edu/data/com-DBLP.html) ($N_{LCC}=317,080$, $E_{LCC}1,049,866$).
> >
> > We trained our model in the full-batch setting (i.e., no node batching or subsampling) and on commodity GPUs (1080Ti). We present the results in the following table and added them to Table 1 of the revised manuscript. **Bold** / *italic* indicate best / second best scores.
> >
> > ||Ward|UF|HGHC|RGHC|Avg. lk.|FPH||Ward|UF|HGHC|RGHC|Avg. lk.|FPH|
> > |-|-|-|-|-|-|-|-|-|-|-|-|-|-|
> > |ogbn-arxiv|22,870|52,666|22,076|25,573|*20,760*|**14,796**||*36.77*|24.75|25.79|23.88|33.46|**38.79**|
> > |ogbl-collab|*13,835*|91,807|34,934|21,057|15,714|**13,661**||45.33|27.90|24.80|32.25|*45.40*|**48.02**|
> > |DBLP|**31,138**|148,439|98,910|44,424|36,463|*31,406*||38.26|20.21|14.70|27.82|38.97|**41.33**|
> >
> > **Table E**: Hierarchical clustering results on additional datasets. Columns 2-7 show Dasgupta results and columns 8-13 show TSD results. We use `n'=512` internal nodes. Note that the HypHC could not run on these datasets because the code constructs a dense $n\times n$ matrix, which lead to OOM. **Bold** / *italic* indicate best / second best results.
> >
> > Our method performs best on all datasets and metrics except one case (DBLP Dasgupta), where it scores a close second place. This highlights the scalability of our method and further emphasizes the quality of our method's discovered hierarchies.
> >
> > |Model|ogbn-arxiv|DBLP|
> > |-|-|-|
> > |Avg.|0.216|0.526|
> > |RGHC|0.286|0.510|
> > |HGHC|0.290|0.408|
> > |Ward|**0.411**|0.591|
> > |UF|0.254|*0.598*|
> > |HypHC|OOM|OOM|
> > |Louvain|0.395|0.558|
> > |FPH|0.251|0.560|
> > |FPH (Louv.)|0.399|0.564|
> > |FPH (Ward)|*0.401*|**0.604**|
> >
> > **Table F**: NMI results on the new datasets with node labels.
> >
> > Here, FPH (Louv.) and FPH (Ward) are our model initialized from Louvain or Ward, respectively. We see that FPH can improve the performance even of strong baselines when used as initialization; with this, FPH scores best or second-best on both new datasets. These results are displayed in the appendix (Table 12) of the revised manuscript.
> >
> > ### Summary
> > We thank the reviewer for their great suggestions on how to improve the experimental section of our paper. We believe that these additional results (three additional datasets, an additional baseline, and results for $d=128$ DeepWalk embeddings) further demonstrate the quality, usefulness and flexibility of our model's discovered hierarchies and hope that you agree. Please let us know in case you have any open questions regarding our work and/or this response.

---

> > > ### Comment · Reviewer_knNt · 2021-11-23
> > > **Fantastic**
> > >
> > > Thank you very much for adding so much content in such a short time. I raise my score and encourage the [S]ACs to accept.

---

### Author Response · Authors · 2021-11-23
**Response to all reviewers (1/2)**

We thank all reviewers for their helpful and constructive comments. In this message we summarize the results of the additional experiments we conducted. We added these also to the revised manuscript; **most new results are in the appendix**, and some were added to Tables 1, 2, and 3 directly. We are happy to move some new results from the appendix into the main text if the reviewers do not object.

### Additional datasets
To highlight our model's performance and scalability and based on Reviewer knNt's suggestion, we add three additional large datasets. These are:
* `ogbn-arxiv` ($N_{LCC}=169,343$, $E_{LCC}=1,157,799$),
* `ogbl-collab` ($N_{LCC}=232,865$, $E_{LCC}=961,883$), and
* `DBLP` [[Yang & Leskovec, 2015]](https://snap.stanford.edu/data/com-DBLP.html) ($N_{LCC}=317,080$, $E_{LCC}1,049,866$).

(Since we select the largest connected component (LCC), we report LCC statistics here). Importantly, we train our model in a **full-batch manner** (i.e., without subsampling) on a single commodity GPU. The results are summarized in Table A. We added these results to Table 1 of the revised manuscript. If the reviewers have objections, we are happy to instead move them to the appendix.

| |Ward|UF|HGHC|RGHC|Avg. lk.|FPH||Ward|UF|HGHC|RGHC|Avg. lk.|FPH |
|-|-|-|-|-|-|-|-|-|-|-|-|-|-|
| ogbn-arxiv|22,870|52,666|22,076|24,077|*20,760*|**14,796**||*36.77*|24.75|26.05|25.21|33.46|**38.79** |
| ogbl-collab|*13,835*|91,807|34,934|21,057|15,714|**13,661**||45.33|27.90|24.80|34.07|*45.40*|**48.02** |
| DBLP|**31,138**|148,439|98,384|44,424|36,463|*31,406*||38.26|20.21|15.96|27.82|38.97|**41.33** |

**Table A**: Hierarchical clustering results on additional datasets. Columns 2-7 show Dasgupta results and columns 8-13 show TSD results. We use `n'=512` internal nodes. Note that the HypHC could not run on these datasets because the code constructs a dense $n\times n$ matrix, which lead to OOM. **Bold** / *italic* indicate best / second best results.

**Our method performs best** on all datasets and metrics except one case (DBLP Dasgupta), where it scores a close second place. This highlights the scalability of our method and further emphasizes the quality of our method's discovered hierarchies.

In addition, we also evaluated the models on ogbn-arxiv and DBLP, i.e. those among the new datasets which have node labels. We summarize the results in the following table, which also contains results for the new Louvain baseline (see "additional baselines"). The table is also in the appendix of the revised manuscript (Table 12); if the reviewers agree, we will replace Table 2 with this table in the camera-ready version.

|Model|Citeseer|Cora|Polblogs|ogbn-arxiv|DBLP|
|-|-|-|-|-|-|
|Avg.|0.367|0.420|0.507|0.216|0.526|
|RGHC|0.281|0.400|**0.730**|0.286|0.510|
|HGHC|0.365|0.379|0.177|0.290|0.408|
|Ward|0.368|0.504|0.702|**0.411**|0.591|
|UF|0.347|0.428|0.676|0.254|*0.598*|
|HypHC|0.270|0.121|0.691|OOM|OOM|
|Louvain|0.329|0.500|0.640|0.395|0.558|
|FPH|**0.398**|0.462|0.680|0.251|0.560|
|FPH (Louv.)|0.380|*0.507*|0.614|0.399|0.564|
|FPH (Ward)|*0.393*|**0.516**|*0.708*|*0.401*|**0.604**|

**Table B**: NMI results on additional datasets and baselines.

Here, FPH (Louv.) and FPH (Ward) are our model initialized from Louvain or Ward, respectively. We see that FPH can improve the performance even of strong baselines when used as initialization; FPH scores best or second-best on each dataset.

### Additional baseline
With the [Louvain algorithm [Blondel et al. 2008]](https://iopscience.iop.org/article/10.1088/1742-5468/2008/10/P10008) we add an additional hierarchical graph clustering baseline. The results are in the appendix (Tables 9, 12), and a subset is in the main text (Tables 2, 3) of the revised manuscript. We show the Louvain results here for convenience. In parentheses we show for comparison our model's results and bold the best score. Note here we include results on three additional datasets (see "Scalability / new datasets" for info).

|Dataset|Dasgupta; lower is better|Norm. TSD; larger is better|
|-|-|-|
|CoraML|336.86 (**269.50**)|57.51 (**58.62**)|
|Citeseer|178.23 (**83.59** )|68.45 (**69.95**)|
|Polblogs|443.48 (**295.44**)|25.93 (**28.56**)|
|Brain|777.14 (**461.17**)|29.28 (**32.64**)|
|Genes|247.26 (**184.35**)|67.47 (**67.65**)|
|WikiPhysics|986.32 (**543.95**)|46.03 (**50.00**)|
|Openflight|633.66 (**348.78**)|51.51 (**57.56**)|
|ogbn-arxiv|31,655 (**14,796**)|37.75 (**38.79**)|
|ogbl-collab|20,664 (**13,661**)|46.12 (**48.02**)|
|DBLP|40,744 (**32,137**)|40.92 (**41.33**)|

**Table C**: Louvain hierarchical clustering results. In parentheses we show FPH's results and we bold the best scores.

The Louvain baseline scores quite competitively overall but our method outperforms it on all of the datasets and clustering metrics. Still, we believe that these results are a valuable addition to the paper, and if the reviewers agree we will move these results into Table 1 in the main text for the camera-ready version.

---

> ### Author Response · Authors · 2021-11-23
> **Response to all reviewers (2/2)**
>
> The comparison to Louvain w.r.t. NMI is shown in Table B of this response. For convenience, we show here only Louvain and FPH / FPH (Louv.):
>
> |Dataset|NMI; higher is better|
> |-|-|
> |Citeseer|0.329  (**0.398**)|
> |Cora|0.500  (0.462 / **0.507**)|
> |Polblogs|0.640  (**0.680**)|
> |ogbn-arxiv|0.395  (0.251 / **0.399**)|
> |DBLP|0.558    (**0.561**)|
> |HSBM (`n=100`, L3)|0.795  (**0.829**)|
> |HSBM (`n=1000`, L3)|0.993  (**0.994**)|
>
> **Table D**: Louvain NMI results. Note again the additional datasets (ogbn-arxiv and DBLP; ogbl-collab does not have node labels).
>
> Here, our method outperforms Louvain in 5/7 cases, again highlighting the quality of FPH's learned hierarchies. On ogbn-arxiv, our method suffers from a poor initialization by average linkage (0.216). The second number (0.399) shows our method's results when we initialize from Louvain's solution. Similarly, FPH improves Louvain's solution on Cora-ML. This shows that our method can improve results of very strong baselines to further increase the quality of the learned hierarchies, which none of the baselines can easily do.
>
> ### DeepWalk embeddings with $d=128$
> To investigate concerns by Reviewer knNt that the DeepWalk embeddings were not expressive enough for the baselines, we conducted additional experiments with $d=128$ dimensions. The results are in the appendix of the revised manuscript; for convenience, we also display them here:
>
> ||Ward|UF|HypHC|HGHC|RGHC||Ward|UF|HypHC|HGHC|RGHC|
> |-|-|-|-|-|-|-|-|-|-|-|-|
> |Brain|**596.73**|938.49|**568.18**|894.87|650.53||**32.43**|26.28|**17.68**|17.31|16.54|
> |OpenFlight|416.05|643.45|**423.80**|**477.51**|**469.98**||**55.59**|49.88|**40.06**|47.52|**45.84**|
> |Genes|221.76|258.21|**467.12**|482.11|444.82||**66.87**|**63.73**|**23.94**|50.59|40.98|
> |Citeseer|105.12|280.66|271.80|224.63|**200.91**||69.28|62.95|31.74|52.53|47.66|
> |Cora-ML|301.47|673.27|**441.21**|516.87|499.49||**57.22**|**47.9**6|29.10|42.10|35.19|
> |PolBlogs|383.51|726.34|334.69|428.52|**376.94**||27.01|10.73|21.60|20.30|**20.78**|
> |WikiPhysics|808.87|958.20|**701.14**|919.27|790.39||**45.54**|41.55|**33.85**|34.51|36.51|
> |ogbn-arxiv|**22,046**|64,950|OOM|37,177|26,286||**37.43**|**26.22**|OOM|17.55|25.20|
> |ogbl-collab|14,834|101,562|OOM|112,048|**17,964**||45.20|**30.50**|OOM|11.11|**37.73**|
> |DBLP|33,349|160,742|OOM|171,975|**41,796**||**38.87**|**22.62**|OOM|5.61|**29.9**|
>
> **Table E**:  Hierarchical clustering results for baselines using DW and $d=128$. **Bold numbers** indicate improved performance for $d=128$. Note that these results include three additional datasets (see "Additional datasets").
>
>  Increasing the embedding dimension did not consistently improve the results for any of the respective baselines; for some baselines (UF, HGHC), results are worse for almost all datasets. In total, we observe an improvement in 33/94 cases, i.e. about one third. We hypothesize that this might be because some of the baselines suffer from the curse of dimensionality, where the Euclidean distances (between the DeepWalk embeddings) tend to become more similar for higher dimensions. Importantly, **none of the baselines outperform FPH** in the $d=128$ scenario. We have added a pointer to this table to Section 4.1 of the revised manuscript.
>
>
>
> ### Summary
> These additional results highlight that **(i)** FPH outperforms the diverse and strong baselines also on three additional datasets; **(ii)** FPH outperforms the additional Louvain baseline in all cases on hierarchical clustering; **(iii)** increasing DeepWalk dimension to $d=128$ does not improve results consistently. If there are any remaining or follow-up questions, we are happy to provide further information.

---

### Decision · Program_Chairs · 2022-01-20

**Decision:**

Accept (Poster)

**Comment:**

The authors introduce a novel probabilistic hierarchical clustering method for graphs. In particular they design an end-to-end gradient-based learning to optimize the Dasgupta cost and Tree Sampling Divergence cost at the same time.

Overall the paper presents solid results both from a theoretical and experimental perspective so I think it is a good fit for the conference and I suggest accepting it.